# Hippocampal damage disrupts the latent decision-making processes underlying approach-avoidance conflict processing in humans

Willem Le Duc[1], Christopher R. Butler[2,3], Georgios P. D. Argyropoulos[4], Sonja Chu[1], Cendri Hutcherson[1,5,6], Anthony C. Ruocco[1,5], Rutsuko Ito[1,5,7], Andy C. H. Lee [1,5,8]*

1 Department of Psychological Clinical Science, University of Toronto, Toronto, Canada, 2 Department of Brain Sciences, Imperial College London, London, United Kingdom, 3 The George Institute for Global Health, London, United Kingdom, 4 Department of Psychology, University of Stirling, Stirling, United Kingdom, 5 Department of Psychology (Scarborough), University of Toronto, Toronto, Canada, 6 Rotman School of Management, University of Toronto, Toronto, Canada, 7 Department of Cell & Systems Biology, University of Toronto, Toronto, Canada, 8 Rotman Research Institute, Baycrest Centre, Toronto, Canada

* andych.lee@utoronto.ca

**Data Availability Statement:** Data are available from the University of Toronto Dataverse (https://doi.org/10.5683/SP3/C4GWZU).

## Abstract

Rodent and human data implicate the hippocampus in the arbitration of approach-avoidance conflict (AAC), which arises when an organism is confronted with a stimulus associated simultaneously with reward and punishment. Yet, the precise contributions of this structure are underexplored, particularly with respect to the decision-making processes involved. We assessed humans with hippocampal damage and matched neurologically healthy controls on a computerized AAC paradigm in which participants first learned whether individual visual images were associated with the reward or loss of game points and were then asked to approach or avoid pairs of stimuli with non-conflicting or conflicting valences. To assess hippocampal involvement more broadly in response conflict, we also administered a Stroop and a Go/No-go task. On the AAC paradigm, following similar learning outcomes in individuals with hippocampal damage and matched controls, both participant groups approached positive and negative image pairs at the same rate but critically, those with hippocampal damage approached conflict pairs more often than controls. Choice and response AAC data were interrogated using the hierarchical drift diffusion model, which revealed that, compared to controls, individuals with hippocampal damage were more biased towards approach, required less evidence to make a decision during conflict trials, and were slower to accumulate evidence towards avoidance when confronted with conflicting image pairs. No significant differences were found between groups in performance accuracy or response time on the response conflict tasks. Taken together, these findings demonstrate the importance of the hippocampus to the evidence accumulation processes supporting value-based decision-making under motivational conflict.

**Funding:** This research was supported by a project grant from the Canadian Institutes of Health Research (https://cihr-irsc.gc.ca) to R.I. and A.C.H.L. (#190154), a University of Toronto Scarborough (https://www.utsc.utoronto.ca/research) International Research Collaboration Fund grant to A.C.H.L., a Medical Research Council (https://www.ukri.org/councils/mrc/) Clinician Scientist Fellowship to C.R.B. (MR/K-010395/1), as well as a Canada Research Chair Award from the Government of Canada (https://www.chairs-chaires.gc.ca/) to C.H. The funders had no role in study design, data collection and analysis, decision to publish, or preparation of the manuscript.

**Competing interests:** The authors have declared that no competing interests exist.

**Abbreviations:** AAC, approach-avoidance conflict; aHPC, anterior hippocampus; Amyg, amygdala; aLE, autoimmune limbic encephalitis; DH, display height; DIC, deviance information criterion; D and P, Doors and People; EMM, estimated marginal mean; ERC, entorhinal cortex; hDDM, hierarchical drift diffusion model; HPC, hippocampus; HSD, honestly significant difference; LMM, linear mixed model; MC, Monte Carlo; MoCA, Montreal Cognitive Assessment; MTLE, medial temporal lobe epilepsy; PHC, parahippocampal cortex; pHPC, posterior hippocampus; PRC, perirhinal cortex; RCFT, Rey Complex Figure Task; VGKC, voltage-gated potassium channel; vHPC, ventral hippocampus; VOSP, Visual Object Spatial Perception Battery; WASI-II, Wechsler Abbreviated Scale of Intelligence, Second Edition; WMS-III, Wechsler Memory Scale, Third Edition; WMS-IV, Wechsler Memory Scale, Fourth Edition.

## Introduction

Approach-avoidance conflict (AAC) arises when potential outcomes of reward and punishment are experienced simultaneously, leading to competing tendencies to engage or retreat [1]. In nonhuman animals, this dilemma is classically illustrated by the prey animal who, in deciding whether to forage for food, must balance the need for resources with the possibility of being exposed to predators. Successful AAC resolution is essential to survival, and dysregulation of approach and avoidance tendencies is suggested to be a characteristic of various mental health disorders [2–5].

A substantial body of rodent research has identified the ventral hippocampus (vHPC) as a key region in the arbitration of AAC [6]. Specifically, gross vHPC damage or inhibition has been shown to increase approach responses to motivationally conflicting stimuli [7], while subfield-specific inactivation can differentially impact approach or avoidance behavior [8,9]. Complementing this work, excitotoxic lesions to the HPC in nonhuman primates, impacting both the anterior HPC (aHPC), the primate homologue of the rodent vHPC, as well as the posterior HPC, have been demonstrated to facilitate the retrieval of reward located near a potential predator [10]. Corresponding human evidence comes primarily from neuroimaging studies that have reported greater activity in the aHPC during high AAC [11,12]. Human HPC dysfunction has also been associated with a greater propensity to approach reward in the presence of threat, although these findings are limited by the use of a paradigm with hippocampal-dependent spatial navigation demands [11] and/or assessment of a single focal bilateral HPC case [13].

Crucially, although the involvement of the HPC in AAC is clear, the aforementioned work has typically focused on behavioral measures (e.g., exploration time, number/proportion of approach responses, response latency) that provide limited insight into the latent computational processes that underpin the observed behavior. Thus, the precise contributions of this structure remain opaque and it is unknown whether the involvement of the HPC in AAC pertains primarily to its role in mnemonic processing or to decision-making processes per se. For example, greater approach behavior under motivational conflict following gross HPC damage may reflect disrupted retrieval of conflicting stimulus valences. Alternatively, lesions to the HPC may alter how evidence is used to guide decision-making, for example, by decreasing attention to negative outcomes and slowing the accumulation of evidence in support of avoidance, and/or decreasing response caution by reducing the amount of evidence necessary to make a decision. Computational models that incorporate choice and response latency data offer a compelling means to address this issue but, although these methods are being applied increasingly to the study of AAC [3,5,14–17], there has been very limited work on the HPC particularly in conjunction with brain lesion cases.

To this end, we recruited a group of 8 individuals with focal hippocampal damage and 25 neurologically healthy controls (see Table 1 for medial temporal lobe structure volumes of hippocampal damage participants and Table 2 for background neuropsychology) and administered a computerized AAC paradigm adapted from previous fMRI work [12], in which participants first learned the reward/punishment outcomes of individual visual images and were then asked to approach or avoid the same items presented as motivationally conflicting or non-conflicting pairs (Fig 1). Two versions of this task, each employing scene or object images, were used to examine the possibility that, in keeping with its role in spatial cognition, the involvement of the HPC in AAC is restricted to spatial/contextual information [18,19]. Besides inspecting standard behavioral indices of AAC decision-making, we employed a Bayesian implementation of the drift diffusion model, the hierarchical drift diffusion model (hDDM) [20,21], which allowed us to investigate the impact of HPC damage on estimates of

**Table 1. Hippocampal damage participant volume differences, expressed as Z-scores, for individual medial temporal lobe regions compared to age-matched controls (mean = 61.18 years old (SD = 8.94); 20 M:8 F).** Data for autoimmune limbic encephalitis (aLE) patients and controls are taken from [45]. Medial temporal lobe epilepsy (MTLE) patient volumetrics were derived using the same methodology described in [45]. In brief, the HPC and Amyg were manually delineated in native space using guidelines described in [77], while PRC, ERC, and PHC were manually delineated in native space using guidelines described in [78]. The HPC was split into anterior (aHPC) and posterior (pHPC) portions, with the aHPC comprising the HPC head extending from the most anterior coronal slice to the first appearance of the uncal apex, and the pHPC comprised the HPC body and tail. All volumes were corrected for intracranial volume prior to Z-score transformation.

| Participant | Aetiology | Left Hemisphere | | | | | | | Right Hemisphere | | | | | | |
|---|---|---|---|---|---|---|---|---|---|---|---|---|---|---|---|
| | | aHPC | pHPC | HPC | Amyg | PRC | ERC | PHC | aHPC | pHPC | HPC | Amyg | PRC | ERC | PHC |
| CHPA | aLE | −1.87 | −2.59 | −3.54 | 2.68 | −1.09 | −1.90 | 0.32 | −2.40 | −1.07 | −2.34 | 2.66 | −1.22 | −1.21 | 0.00 |
| COSA | aLE | −3.32 | −1.57 | −3.45 | −0.01 | −0.98 | −1.67 | 0.23 | −5.59 | −3.74 | −5.54 | −0.42 | −0.83 | 0.14 | 1.09 |
| DAFI | aLE | −3.84 | −0.35 | −2.90 | −0.11 | −0.39 | −0.09 | −0.45 | −4.48 | 1.62 | −1.76 | 0.08 | 0.88 | −1.23 | −0.15 |
| JODA | aLE | −5.99 | −3.98 | −7.10 | −0.64 | −0.30 | 0.64 | −0.29 | −6.01 | −2.61 | −5.12 | 0.00 | −0.85 | −1.17 | −0.22 |
| JORO | aLE | −2.14 | −2.34 | −3.17 | −0.94 | 1.28 | 0.59 | −1.03 | −1.31 | −1.28 | −1.75 | −1.50 | 0.44 | −0.63 | −0.27 |
| KEHA | aLE | −1.86 | −3.29 | −3.65 | −2.14 | −0.74 | −0.67 | −0.91 | −1.04 | −1.70 | −1.85 | −1.41 | −1.25 | −1.09 | −1.03 |
| PAFO | aLE | −2.33 | −4.30 | −3.57 | −1.03 | 0.00 | −1.01 | −0.05 | −3.35 | −4.88 | −4.56 | −0.44 | 0.37 | −0.02 | −0.67 |
| MIPI | MTLE | −2.47 | −3.89 | −4.58 | 1.61 | −1.13 | −0.98 | 0.07 | −2.42 | −3.69 | −3.61 | 1.13 | −0.47 | −0.31 | 0.02 |
| **Mean** | | **−2.98** | **−2.79** | **−4.00** | **−0.07** | **−0.42** | **−0.63** | **−0.26** | **−3.33** | **−2.17** | **−3.31** | **0.01** | **−0.37** | **−0.69** | **−0.15** |
| **SD** | | **1.40** | **1.35** | **1.34** | **1.54** | **0.80** | **0.95** | **0.50** | **1.88** | **2.03** | **1.59** | **1.36** | **0.82** | **0.56** | **0.62** |

aHPC, anterior hippocampus; Amyg, amygdala; aLE, autoimmune limbic encephalitis; ERC, entorhinal cortex; HPC, hippocampus; MTLE, medial temporal lobe epilepsy; pHPC, posterior hippocampus.

baseline propensity towards approach or avoidance (bias), rates of evidence accumulation towards approach or avoidance (drift rate), the amount of evidence needed for a given decision (threshold), and the recruitment of non-decision cognitive processes (i.e., non-decision time). Lastly, in light of work demonstrating impaired responding under response conflict in epilepsy patients with HPC sclerosis [22,23], participants were also administered computerized classic Stroop and Go/No-Go tasks. This allowed us to examine whether the HPC plays a wider role in conflict processing beyond value-based decision-making.

## Results

### Approach-avoidance tasks

**Linear mixed model analyses of choice and response time data.** Learning Phase–Participants first learned the valences of 4 scene (Scene task) or 4 object (Object task) stimuli over 120 trials by making approach or avoid key presses to individually presented stimuli. Approaching a positive stimulus led to the award of game points, whereas approaching a negative stimulus led to the loss of game points. An avoid response had no impact on game points regardless of stimulus valence. The proportion of correct responses was analyzed using a linear mixed model (LMM) with fixed effects of group (Hippocampal Damage versus Control), valence (Positive versus Negative), block (1 to 10, with 12 trials [i.e., 3 repetitions of each stimulus] per block), stimuli (Object versus Scene), and the interactions between them as predictors. We additionally modeled random intercepts and slopes for valence per participant. The selected model's formula was: accuracy ~ group * valence * block * stimuli + (1 + valence | participant). Table 3A summarizes the selected model's outputs, as well as post hoc estimated marginal mean (EMM) comparisons (adjusted for multiple comparisons using Tukey's honestly significant difference (HSD) method). To correct for multiple LMMs being conducted in this study (7 in total), all $p$-values for this model and subsequent LMMs were additionally adjusted with a Bonferroni correction ($p_{corr} = p \times 7$).

**Table 2. Mean hippocampal damage participant and control test scores (raw or %) for standard neuropsychological tests including the MoCA [63], the WMS-III [64] and WMS-IV [65], the D and P [66], the RCFT [67], the WASI-II [62], and the VOSP [68].** Performance qualitative descriptors are taken from published test norms.

| Neuropsychological Test | Hippocampal Damage $N$ | Score (SD) | Qualitative Descriptor | Control $N$ | Score (SD) | Qualitative Descriptor |
|---|---|---|---|---|---|---|
| MoCA (/30) | 0 | N/A | N/A | 25 | 26.41 (2.04) | Pass |
| WASI-II Matrix Reasoning (/30) | 7 | 18.13 (3.48) | Average | 24 | 17.38 (3.49) | Average |
| WASI-II Similarities (/45) | 7 | 35.86 (4.45) | High Average | 24 | 33.95 (4.22) | High Average |
| WASI-II Vocabulary (/59) | 8 | 44.38 (6.61) | High Average | 25 | 42.64 (4.76) | High Average |
| WMS-III LM Units IR (%) | 8 | 41.84 (13.97) | Average | 0 | N/A | N/A |
| WMS-IV LM IR (%) | 0 | N/A | N/A | 21 | 68.23 (12.87) | High Average |
| WMS-III LM Units DR (%) | 8 | 25.33 (11.92) | Low Average | 0 | N/A | N/A |
| WMS-IV LM DR (%) | 0 | N/A | N/A | 21 | 52.68 (16.72) | High Average |
| WMS-III Word List IR (%) | 8 | 52.38 (17.40) | Average | 0 | N/A | N/A |
| WMS-IV VPA IR (%) | 0 | N/A | N/A | 22 | 74.36 (22.04) | High Average |
| WMS-III Word List DR (%) | 8 | 38.33 (24.14) | Average | 0 | N/A | N/A |
| WMS-IV VPA DR (%) | 0 | N/A | N/A | 22 | 68.89 (23.39) | High Average |
| WMS-III Word List Recognition (%) | 8 | 81.02 (26.53) | Average | 0 | N/A | N/A |
| WMS-IV VPA Recognition(%) | 0 | N/A | N/A | 22 | 96.10 (4.17) | Average |
| WMS-III Digit Span (/50) | 8 | 19.00 (5.01) | High Average | 0 | N/A | N/A |
| WMS-IV Symbol Span (/50) | 0 | N/A | N/A | 25 | 18.77 (5.33) | Average |
| RCFT Copy (/36) | 7 | 34.71 (1.89) | WNL | 25 | 31.64 (2.17) | WNL |
| RCFT IR (/36) | 7 | 20.29 (6.80) | Average | 25 | 18.00 (5.64) | High Average |
| RCFT DR (/36) | 7 | 20.79 (5.86) | Average | 25 | 18.68 (5.92) | High Average |
| D and P Doors Recognition (/24) | 8 | 18.50 (2.67) | Average | 24 | 15.33 (2.78) | Average |
| D and P People IR (/36) | 8 | 20.13 (6.94) | Low Average | 25 | 27.18 (4.77) | Average |
| D and P People Forgetting (/12) | 8 | 3.00 (3.16) | Average | 25 | 1.59 (1.82) | Average |
| D and P Names Recognition (/24) | 8 | 14.50 (4.72) | Average | 24 | 18.10 (2.91) | High Average |
| D and P Shapes IR (/36) | 8 | 31.75 (3.37) | Average | 21 | 31.06 (7.50) | High Average |
| D and P Shapes Forgetting (/12) | 8 | 0.88 (3.09) | Average | 21 | 0.22 (0.55) | Average |
| VOSP Dot Counting (/10) | 8 | 10.00 (0.00) | WNL | 22 | 9.95 (0.23) | WNL |
| VOSP Position Discrimination (/20) | 8 | 19.75 (0.46) | WNL | 22 | 19.47 (0.77) | WNL |
| VOSP Cube Analysis (/10) | 8 | 9.75 (0.46) | WNL | 22 | 8.47 (2.72) | WNL |

DC, Dot Counting; D and P, Doors and People; DR, Delayed Recall; IR, Immediate Recall; LM, Logical Memory; MoCA, Montreal Cognitive Assessment; PD, Position Discrimination; RCFT, Rey Complex Figure Task; VOSP, Visual Object Spatial Perception Battery; VPA, Verbal Paired Associates; WASI-II, Wechsler Abbreviated Scale of Intelligence, Second Edition; WMS-III, Wechsler Memory Scale, Third Edition; WMS-IV, Wechsler Memory Scale, Fourth Edition; WNL, Within Normal Limits.

Based on our predictions, post hoc pairwise comparisons of EMMs, selected a priori, were performed to probe relevant main and interaction effects. Specifically, we sought to find out whether: (1) participants showed significantly improved accuracy from Block 1 to Block 10; (2) both groups showed similar accuracy at Block 10 on both tasks; and (3) performance was similar for negative and positive stimuli for the Scene and Object tasks at Block 10. We found a main effect of block ($p_{corr} < 0.001$) and a significant block-by-group ($p_{corr} = 0.007$) interaction. There were also trends for the block-by-group-by-stimuli ($p_{corr} = 0.063$) and valence-by-block-by-stimuli ($p_{corr} < 0.091$) interactions, but no significant interaction between valence, block, and group ($p_{corr} = 1.000$). Comparing EMMs at Block 1 and at Block 10 revealed, as expected, that the proportion of accurate responses increased significantly from Block 1 to Block 10 ($p_{corr} < 0.001$; Fig 2A). When compared at Block 10, both groups had similar accuracies on both the Scene ($p_{corr} = 1.000$) and Object tasks ($p_{corr} = 1.000$). Collapsing across groups, there

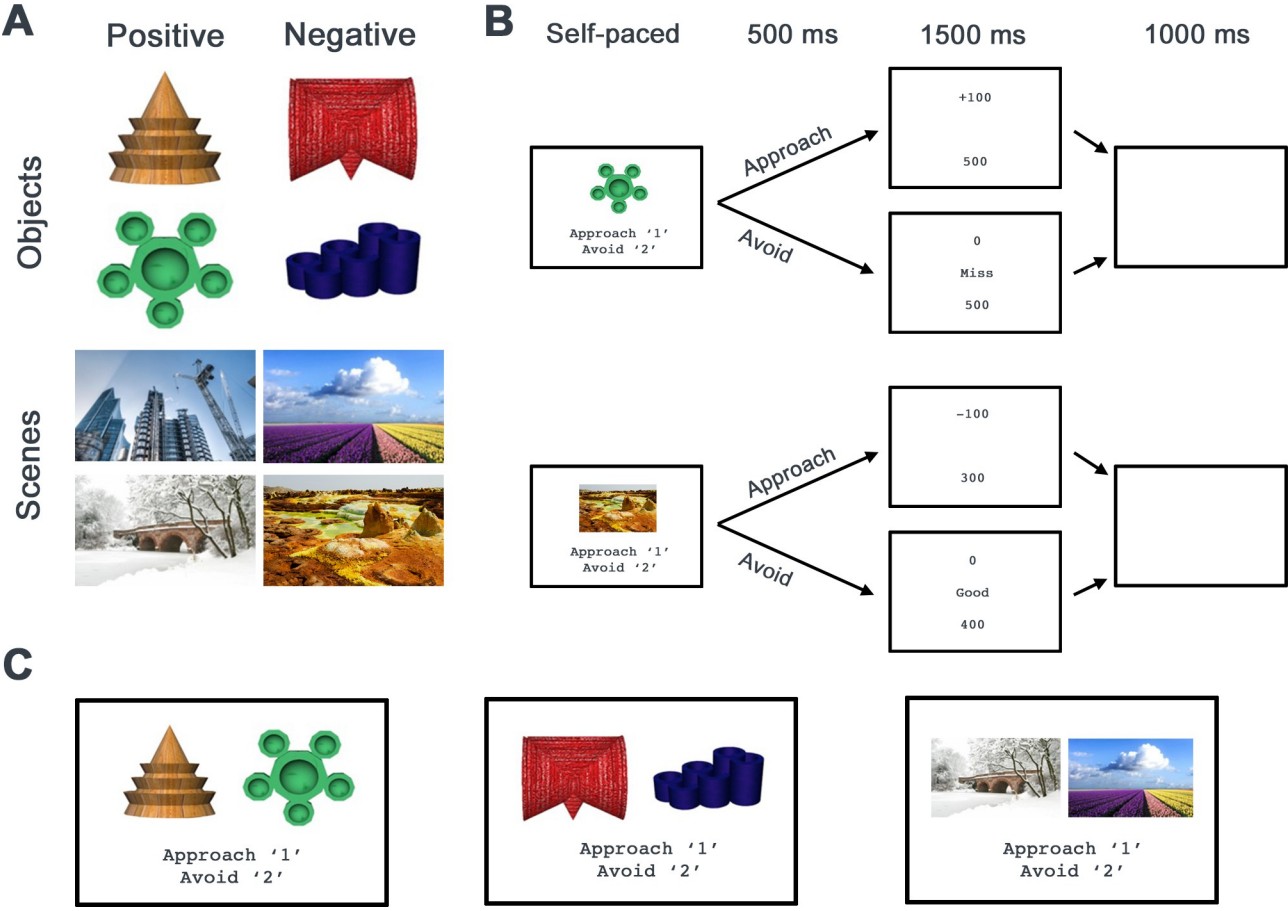

**Fig 1.** (A) The Object and Scene AAC tasks each involved 4 unique stimuli, with 2 assigned to be positive and 2 assigned to be negative in valence (note: to comply with license restrictions, one of the scene task stimuli (bottom right) has been replaced with a similar image for display purposes). (B) During an initial learning phase, participants learned to approach or avoid these stimuli in order to gain or avoid the loss of game points, respectively. An example positive trial from the Object AAC task (top) and a negative trial from the Scene AAC task (bottom) are depicted. Participants were presented with a feedback screen after each response, showing the outcome of their response and their total accumulated points. (C) During a subsequent decision phase, participants were then asked to approach or avoid pairs of stimuli in the absence of feedback, with each pair composed of stimuli with non-conflicting positive, non-conflicting negative, or conflicting valences. Examples of Object No-Conflict Positive (left), Object No-Conflict Negative (middle), and Scene Conflict (right) pairs from the Object and Scene AAC tasks are shown. AAC, approach-avoidance conflict.

was no significant difference in performance at Block 10 between negative and positive stimuli in the Scene task ($p_{corr}$ = 1.000) or Object task ($p_{corr}$ = 1.000). These results indicate that both groups learned the images' valences over the course of learning trials and had achieved comparable knowledge of them by the end of the learning phase. Moreover, learning of negative and positive valenced stimuli was similar.

We next used a LMM to analyze response times to determine whether participants showed increased familiarity with the stimulus images by the end of the learning phase. We used identical fixed predictors to the abovementioned LMM but modeled random intercepts and slopes for stimuli per participant. The formula for the selected model was: RT ~ group * valence * block * stimuli + (1 + stimuli | participant). Table 3B summarizes the selected model's outputs as well as post hoc EMM comparisons (adjusted for multiple comparisons using Tukey's HSD).

As expected, we observed a significant main effect of block ($p_{corr}$ < 0.001). To determine whether this reflected faster response times over the course of the task, we compared EMMs

**Table 3. Object and Scene AAC task learn phase LMM results for (A) Accuracy; and (B) Response time.** All post hoc EMM comparisons are adjusted for multiple comparisons using Tukey's HSD ($p_{HSD}$). To correct for multiple LMMs being conducted in this study (7 in total), all $p$-values have additionally been adjusted with a Bonferroni correction ($p_{corr}$). Significant Bonferroni corrected findings are highlighted in bold (* < 0.05; ** < 0.01, *** < 0.001).

**A. Accuracy**

*Model Summary*

| Predictor | $b$ | 95% CI | $p$ | $p_{corr}$ |
|---|---|---|---|---|
| (Intercept) | 0.85 | 0.82–0.87 | <0.001 | **<0.001**\*\*\* |
| Valence | 0.05 | 0.02–0.08 | <0.001 | **0.002**\*\* |
| Block | 0.02 | 0.02–0.02 | <0.001 | **<0.001**\*\*\* |
| Group | 0.04 | −0.01–0.09 | 0.156 | 1.000 |
| Stimuli | 0.02 | −0.01–0.04 | 0.188 | 1.000 |
| Valence * Block | −0.01 | −0.01 −−0.00 | 0.004 | **0.028**\* |
| Valence * Group | −0.03 | −0.09–0.03 | 0.342 | 1.000 |
| Block * Group | −0.01 | −0.01 −−0.00 | 0.001 | **0.007**\*\* |
| Valence * Stimuli | 0.07 | 0.02–0.11 | 0.007 | **0.049**\* |
| Block * Stimuli | −0.00 | −0.00–0.00 | 0.596 | 1.000 |
| Group * Stimuli | −0.08 | −0.12 −−0.03 | 0.002 | **0.014**\* |
| Valence * Block * Group | 0.00 | −0.00–0.01 | 0.421 | 1.000 |
| Valence * Block * Stimuli | −0.01 | −0.02 −−0.00 | 0.013 | 0.091 |
| Valence * Group * Stimuli | 0.07 | −0.03–0.16 | 0.170 | 1.000 |
| Block * Group * Stimuli | 0.01 | 0.00–0.02 | 0.009 | 0.063 |
| Valence * Block * Group * Stimuli | −0.01 | −0.02–0.01 | 0.284 | 1.000 |
| Random Effects | | | | |
| $\sigma^2$ | 0.04 | | | |
| $\tau_{00\ participant}$ | 0.00 | | | |
| $\tau_{11\ participant*valence}$ | 0.00 | | | |
| $\rho_{01\ participant}$ | -0.93 | | | |
| Intraclass Correlation Coefficient | 0.09 | | | |
| N | 33 | | | |
| Observations | 7,560 | | | |
| Marginal $R^2$ / Conditional $R^2$ | 0.069 / 0.152 | | | |

*Post hoc EMM Comparisons*

| Contrast | Estimate (SE) | $t$ (df) | $p_{HSD}$ | $p_{corr}$ | Cohen's $d$ | 95% CI |
|---|---|---|---|---|---|---|
| Block 1 –Block 10 | −0.17 (0.01) | −23.09 (7,483) | <0.001 | **<0.001**\*\*\* | 0.83 | [0.76, 0.90] |
| Scene Task Block 10: Controls–Hippocampal Damage | 0.01 (0.03) | 0.49 (53.60) | 0.625 | 1.00 | 0.07 | [−0.21, 0.35] |
| Object Task Block 10: Controls–Hippocampal Damage | 0.04 (0.03) | 1.42 (50.20) | 0.163 | 1.00 | 0.19 | [−0.08, 0.47] |
| Scene Task Block 10: Negative–Positive | 0.02 (0.01) | 1.20 (216) | 0.230 | 1.00 | 0.09 | [−0.05, 0.23] |
| Object Task Block 10: Negative–Positive | −0.01 (0.01) | −0.56 (207) | 0.578 | 1.00 | −0.04 | [−0.18, 0.09] |

**B. Response Time**

*Model Summary*

| Predictor | $b$ | 95% CI | $p$ | $p_{corr}$ |
|---|---|---|---|---|
| (Intercept) | 2,564.65 | 2,169.94–2959.36 | <0.001 | **<0.001**\*\*\* |
| Valence | −12.48 | −372.94–347.99 | 0.946 | 1.00 |
| Block | −197.31 | −226.36 −−168.26 | <0.001 | **<0.001**\*\*\* |
| Group | −984.85 | −1,774.27 −−195.43 | 0.014 | 0.098 |
| Stimuli | 10.49 | −439.43–460.42 | 0.964 | 1.00 |
| Valence * Block | 3.35 | −54.75–61.44 | 0.910 | 1.00 |
| Valence * Group | −43.80 | −764.73–677.13 | 0.905 | 1.00 |
| Block * Group | 114.62 | 56.53–172.72 | <0.001 | **<0.001**\*\*\* |

*(Continued)*

**Table 3.** (Continued)

| | | | | |
|---|---|---|---|---|
| Valence * Stimuli | 47.07 | −673.86–768.00 | 0.898 | 1.00 |
| Block * Stimuli | −10.25 | −68.35–47.84 | 0.729 | 1.00 |
| Group * Stimuli | −285.43 | −1,185.28–614.42 | 0.534 | 1.00 |
| Valence * Block * Group | 6.71 | −109.48–122.90 | 0.910 | 1.00 |
| Valence * Block * Stimuli | −8.16 | −124.35–108.03 | 0.891 | 1.00 |
| Valence * Group * Stimuli | −996.89 | −2,438.75–444.97 | 0.175 | 1.00 |
| Block * Group * Stimuli | 72.03 | −44.15–188.22 | 0.224 | 1.00 |
| Valence * Block * Group * Stimuli | 101.03 | −131.35–333.40 | 0.394 | 1.00 |
| Random Effects | | | | |
| $\sigma^2$ | 9,903,338.98 | | | |
| $\tau_{00\ participant}$ | 766,236.94 | | | |
| $\tau_{11\ participant*stimuli}$ | 409,952.93 | | | |
| $\rho_{01}$ | −0.15 | | | |
| Intraclass Correlation Coefficient | 0.08 | | | |
| N | 33 | | | |
| Observations | 7,560 | | | |
| Marginal $R^2$ / Conditional $R^2$ | 0.043 / 0.120 | | | |

***Post hoc EMM Comparisons***

| Contrast | Estimate (SE) | $t$ (df) | $p_{HSD}$ | $p_{corr}$ | Cohen's $d$ | 95% CI |
|---|---|---|---|---|---|---|
| Block 1 –Block 10 | 2,044 (113) | 18.02 (7,485) | <0.001 | **<0.001***** | 0.65 | [0.58, 0.72] |
| Block 10: Controls–Hippocampal Damage | −158 (392) | −0.40 (39.80) | 0.689 | 1.000 | −0.05 | [−0.30, 0.20] |

AAC, approach-avoidance conflict; HSD, honestly significant difference; LMM, linear mixed model.

for response time at Block 1 and Block 10. Indeed, participants responded significantly faster at Block 10 than they did at Block 1 ($p_{corr} < 0.001$; Fig 2B), suggesting they had become more familiar with the task stimuli by the end of the learning phase. A block-by-group interaction was observed ($p_{corr} < 0.001$), although a post hoc comparison revealed that the groups did not differ in their response times at Block 10 ($p_{corr} = 1.000$). There was no significant interaction between valence, block, and group ($p_{corr} = 1.000$) or block, group, and stimuli ($p_{corr} = 1.000$).

**Decision phase**–Following the learning phase, participants made approach or avoid key presses to pairs of scenes (Scene task) or objects (Object task) across 108 trials. Two thirds of the trials contained pairs composed of stimuli with the same valence (No-Conflict Positive and No-Conflict Negative trials) and a third of the trials involved a positive stimulus paired with a negative stimulus (Conflict trials). Participants were told that approaching a Conflict pair would lead to a 50% chance of receiving a gain or loss of game points. We analyzed decision phase approach/avoid responses with a LMM with group (Hippocampal Damage versus Control), condition (No-Conflict Positive versus No-Conflict Negative versus Conflict), and stimuli (Object versus Scene), as well as the interactions between them as fixed effects, with random intercepts per participant. The formula for the selected model was: response ~ group + positive_vs_conflict + negative_vs_conflict + stimuli + group * positive_vs_conflict + group * negative_vs_conflict + group * positive_vs_conflict * stimuli + group * negative_vs_conflict * stimuli + (1 | participant). Table 4A summarizes the selected model's outputs, effect tests for multi-parameter predictors, and post hoc EMM comparisons (adjusted for multiple comparisons using Tukey's HSD).

Multi-parameter tests revealed a significant main effect of condition ($p_{corr} < 0.001$) as well as a significant group-by-condition interaction ($p_{corr} < 0.001$). Probing the main effect of

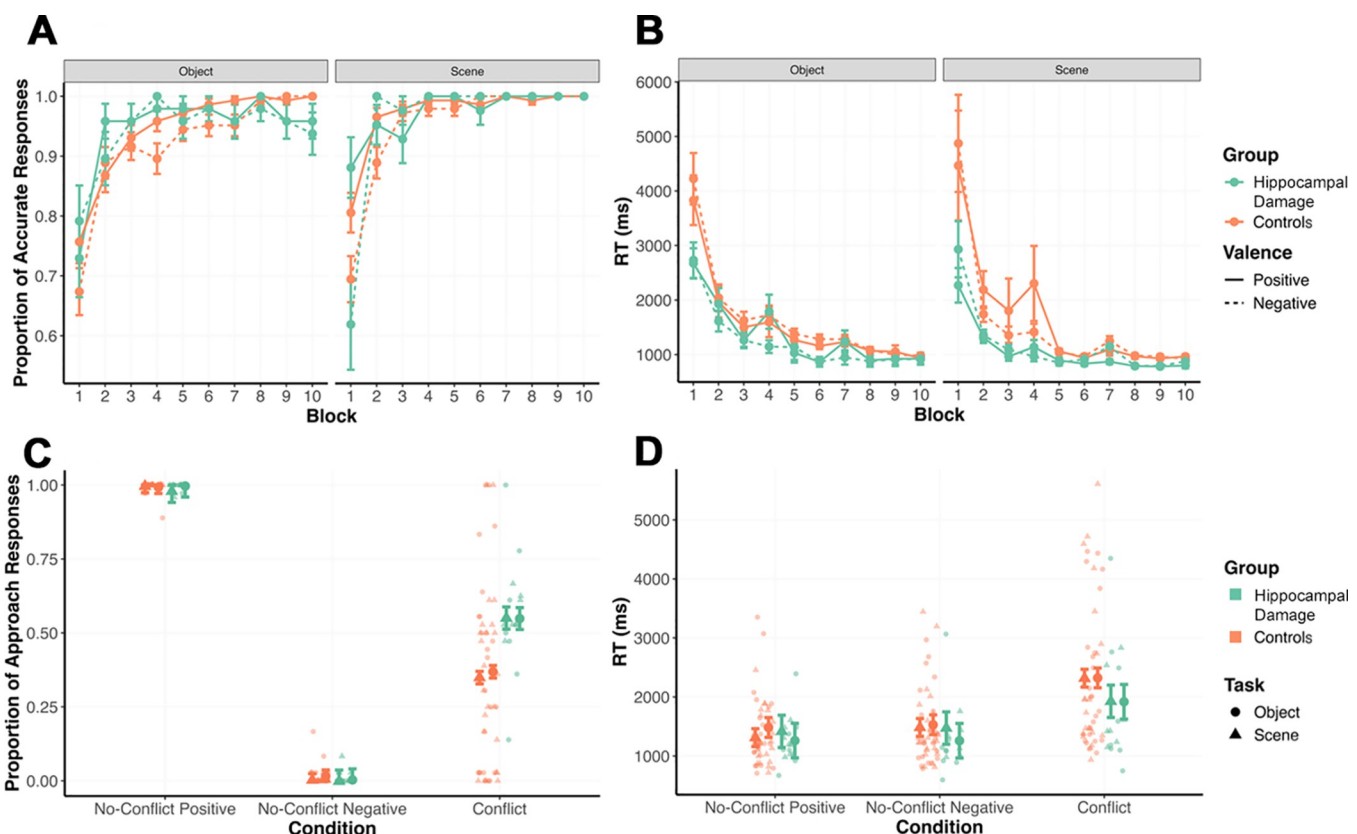

**Fig 2.** Both participant groups successfully acquired the stimulus valences across trial blocks during the learning phase, as reflected in (A) accuracy and (B) response times (RT) (means ± SE). Importantly, the individuals with hippocampal damage and controls demonstrated comparable learning by the final block (all $p_{corr}$ = 1.000). In the decision phase, (C) individuals with hippocampal damage responded similarly to controls on No-Conflict Positive and No-Conflict Negative trials (both $p_{corr}$ = 1.000) but approached significantly more often in response to Conflict image pairs ($p_{corr}$ < 0.001). There were no significant differences in (D) response times between groups (all $p_{corr}$ = 1.000) (individual data points with EMMs ±SE). Underlying data for these figures and associated analyses are available from https://doi.org/10.5683/SP3/C4GWZU. EMM, estimated marginal mean.

condition by comparing EMMs for all 3 conditions to one another revealed that rates of approach response differed significantly across all 3 conditions (all $p_{corr}$ < 0.001), suggesting that participants retained the stimulus identities and their respective valences acquired during the learning phase, with proportions of approach close to 0 for No-Conflict Negative trials, close to 1 for No-Conflict Positive trials, and in between for Conflict trials (Fig 2C). To probe the group-by-condition interaction, we next compared group EMMs for each condition. Groups did not differ significantly in their rates of approach responses on either No-Conflict Positive trials ($p_{corr}$ = 1.000) or No-Conflict Negative ($p_{corr}$ = 1.000), indicating comparable retention of stimulus-valence associations at test. Importantly, consistent with our hypotheses, the hippocampal damage group approached significantly more often than the control group on Conflict trials ($p_{corr}$ < 0.001). Contrary to our hypotheses, however, stimulus type did not interact significantly with group or trial type (both $p$ = 1.000), indicating that the group differences observed in approach behavior on Conflict trials were not specific to either objects or scenes.

Next, we analyzed response times to determine whether the task paradigm successfully elicited AAC and to investigate whether groups differed in the impact of Conflict on their response speed. To this end, we used a LMM with the same fixed and random effects structure

**Table 4. Object and Scene AAC task decision phase LMM results for (A) Proportion of approach responses; and (B) Response time.** Since there were 3 trial types (No-Conflict Positive, No-Conflict Negative, Conflict), condition was coded as a contrast between Conflict and each of the No-Conflict conditions (Positive, Negative). Significant predictors involving these contrasts were then explored further with 2 multi-parameter effects tests, one to assess the significance of the main effect of condition and another the interaction between group and condition. All post hoc EMM comparisons are adjusted for multiple comparisons using Tukey's HSD ($p_{HSD}$). To correct for multiple LMMs being conducted in this study (7 in total), all $p$-values have additionally been adjusted with a Bonferroni correction ($p_{corr}$). Significant Bonferroni corrected findings are highlighted in bold (* < 0.05; ** < 0.01, *** < 0.001).

**A. Proportion of Approach Responses**

*Model Summary*

| Predictor | $b$ | 95% CI | $p$ | $p_{corr}$ |
|---|---|---|---|---|
| (Intercept) | 0.48 | 0.45–0.52 | <0.001 | **<0.001***** |
| Group | 0.06 | −0.02–0.14 | 0.135 | 1.000 |
| Positive | 0.54 | 0.52–0.56 | <0.001 | **<0.001***** |
| Negative | −0.45 | −0.47––0.43 | <0.001 | **<0.001***** |
| Stimuli | −0.01 | −0.02–0.01 | 0.301 | 1.000 |
| Group * Positive | −0.20 | −0.23––0.16 | <0.001 | **<0.001***** |
| Group * Negative | −0.20 | −0.24––0.16 | <0.001 | **<0.001***** |
| Group * Stimuli | 0.00 | −0.03–0.03 | 0.858 | 1.000 |
| Positive * Stimuli | 0.00 | −0.04–0.04 | 0.921 | 1.000 |
| Negative * Stimuli | 0.00 | −0.04–0.04 | 0.990 | 1.000 |
| Group * Positive * Stimuli | −0.04 | −0.12–0.03 | 0.265 | 1.000 |
| Group * Negative * Stimuli | −0.01 | −0.09–0.06 | 0.726 | 1.000 |

Random Effects

| | | | | |
|---|---|---|---|---|
| $\sigma^2$ | 0.07 | | | |
| $\tau_{00\ participant}$ | 0.01 | | | |
| Intraclass Correlation Coefficient | 0.11 | | | |
| N | 33 | | | |
| Observations | 6,804 | | | |
| Marginal $R^2$ / Conditional $R^2$ | 0.665 / 0.701 | | | |

*Multi-Parameter Effect Tests*

| Effect | Predictors | $\Delta_{AIC}$ | $\Delta_{BIC}$ | $\chi^2$(df) | $p$ | $p_{corr}$ |
|---|---|---|---|---|---|---|
| Condition main effect | Positive, Negative | 6,423.80 | 6,410.20 | 6,427.90 (2) | <0.001 | **<0.001***** |
| Group-by-condition interaction | Group * Positive, Group * Negative | 138.30 | 124.70 | 142.34 (2) | <0.001 | **<0.001***** |

*Post hoc EMM Comparisons*

| Contrast | Estimate (SE) | $t$ (df) | $p_{HSD}$ | $p_{corr}$ | Cohen's $d$ | 95% CI |
|---|---|---|---|---|---|---|
| Conflict–Positive | −0.54 (0.01) | −56.28 (6,761) | <0.001 | **<0.001***** | 1.97 | [1.89, 2.04] |
| Conflict–Negative | 0.45 (0.01) | 47.07 (6,761) | <0.001 | **<0.001***** | 1.64 | [1.57, 1.72] |
| Positive–Negative | 0.99 (0.01) | 103.35 (6,761) | <0.001 | **<0.001***** | 3.61 | [3.52, 3.70] |
| Positive: Controls–Hippocampal Damage | 0.01 (0.04) | 0.16 (36.20) | 0.873 | 1.000 | 0.02 | [−0.28, 0.33] |
| Negative: Controls–Hippocampal Damage | 0.01 (0.04) | 0.20 (36.20) | 0.841 | 1.000 | 0.03 | [−0.27, 0.33] |
| Conflict: Controls–Hippocampal Damage | −0.19 (0.04) | −4.68 (36.20) | <0.001 | **<0.001***** | 0.70 | [0.40, 1.00] |

**B. Response Time**

*Model Summary*

| Predictor | $b$ | 95% CI | $p$ | $p_{corr}$ |
|---|---|---|---|---|
| (Intercept) | 1,642.15 | 1,361.11–1,923.20 | <0.001 | **<0.001***** |
| Group | −198.67 | −760.76–363.41 | 0.488 | 1.000 |
| Positive | −753.51 | −876.32––630.71 | <0.001 | **<0.001***** |
| Negative | −685.44 | −808.24––562.64 | <0.001 | **<0.001***** |
| Stimuli | 25.99 | −198.68–250.66 | 0.821 | 1.000 |
| Group * Positive | 339.36 | 93.75–584.96 | 0.007 | **0.049*** |
| Group * Negative | 257.55 | 11.94–503.15 | 0.040 | 0.28 |

*(Continued)*

**Table 4.** (*Continued*)

| | | | | | |
|---|---|---|---|---|---|
| Group * Stimuli | 200.25 | −249.09–649.59 | | 0.382 | 1.000 |
| Positive * Stimuli | −9.83 | −255.43–235.78 | | 0.937 | 1.000 |
| Negative * Stimuli | 82.86 | −162.75–328.46 | | 0.508 | 1.000 |
| Group * Positive * Stimuli | 316.36 | −174.86–807.57 | | 0.207 | 1.000 |
| Group * Negative * Stimuli | 247.51 | −243.70–738.72 | | 0.323 | 1.000 |
| **Random Effects** | | | | | |
| $\sigma^2$ | 3,218,217.71 | | | | |
| $\tau_{00\ participant}$ | 476,036.32 | | | | |
| $\tau_{11\ participant.stimuli}$ | 229,141.91 | | | | |
| $\rho_{01\ participant}$ | −0.19 | | | | |
| Intraclass Correlation Coefficient | 0.14 | | | | |
| N | 33 | | | | |
| Observations | 6,804 | | | | |
| Marginal $R^2$ / Conditional $R^2$ | 0.040 / 0.177 | | | | |

**Multi-Parameter Effect Tests**

| Effect | Predictors | $\Delta_{AIC}$ | $\Delta_{BIC}$ | $\chi^2$(df) | $p$ | $p_{corr}$ |
|---|---|---|---|---|---|---|
| Condition main effect | Positive, Negative | 171 | 157 | 174.98 (2) | <0.001 | **<0.001***\*** |
| Group-by-condition interaction | Group * Positive, Group * Negative | 4 | 10 | 8.00 (2) | 0.018 | 0.126 |

**Post hoc EMM Comparisons**

| Contrast | Estimate (SE) | $t$ (df) | $p_{HSD}$ | $p_{corr}$ | Cohen's $d$ | 95% CI |
|---|---|---|---|---|---|---|
| Conflict–Positive | 753.50 (62.60) | 12.03 (6,733) | <0.001 | **<0.001***\*** | 0.42 | [0.35, 0.49] |
| Conflict–Negative | 685.40 (62.60) | 10.94 (6,733) | <0.001 | **<0.001***\*** | 0.38 | [0.31, 0.45] |
| Positive–Negative | −68.10 (62.60) | −1.09 (6,733) | 0.700 | 1.000 | −0.04 | [−0.11, 0.03] |
| Positive: Controls–Hippocampal Damage | 58.30 (296) | 0.20 (35.20) | 0.845 | 1.000 | 0.03 | [−0.30, 0.37] |
| Negative: Controls–Hippocampal Damage | 140.10 (296) | 0.47 (35.20) | 0.639 | 1.000 | 0.08 | [−0.26, 0.41] |
| Conflict: Controls–Hippocampal Damage | 397.60 (296) | 1.34 (35.20) | 0.188 | 1.000 | 0.22 | [−0.11, 0.56] |

AAC, approach-avoidance conflict; EMM, estimated marginal mean; HSD, honestly significant difference; LMM, linear mixed model.

as that described above, implementing the formula: RT ~ group + conflict_vs_positive + conflict_vs_negative + stimuli + group * conflict_vs_positive + group * conflict_vs_negative + group * conflict_vs_positive * stimuli + group * conflict_vs_negative * stimuli + (1 | participant). Table 4B summarizes the selected model's outputs, effect tests for multi-parameter predictors, and post hoc EMM comparisons (adjusted for multiple comparisons using Tukey's HSD).

A multi-parameter test revealed a main effect of condition on response time ($p_{corr} < 0.001$) and a comparison of EMMs revealed no significant difference between No-Conflict Positive and No-Conflict Negative trials ($p_{corr} = 1.000$). However, Conflict trials had significantly longer response times compared to No-Conflict Positive ($p_{corr} < 0.001$) and No-Conflict Negative trials ($p_{corr} < 0.001$; Fig 2D), suggesting that these trials successfully elicited AAC. We also observed a group-by-condition interaction ($p = 0.018$), although this did not survive Bonferroni correction ($p_{corr} = 0.126$). Exploratory post hoc comparisons of EMMs found similar response times between groups on all 3 conditions (all $p_{corr} \geq 1.000$).

## Hierarchical drift diffusion model analyses

The candidate hDDM model that converged successfully and showed best fit for the decision phase data made separate estimates for each parameter, varying by condition (i.e., drift rate,

threshold, and non-decision time), except bias (deviance information criterion (DIC) = 9848.59; all alternative models DIC > 9862). Bias was modeled collapsed across conditions because it is conceptualized as the starting point for evidence accumulation *before* participants are exposed to the condition of each trial. We also modeled participant-wise estimates for every parameter, except for bias, which we modeled only at the group level to achieve convergence.

**Within-group comparisons.** Parameter estimates largely differed between task conditions as expected. In both groups, model parameter estimates suggested that non-decision times on No-Conflict conditions were likely near-identical (Hippocampal Damage: $P_{Positive > Negative}$ = 0.493; Controls: $P_{Positive > Negative}$ = 0.378; Fig 3A) but were almost certainly longer on Conflict trials (Hippocampal Damage: $P_{Conflict > Positive}$ = 0.984, $P_{Conflict > Negative}$ = 0.982; Controls: $P_{Conflict > Positive}$ > 0.999, $P_{Conflict > Negative}$ > 0.999). Posterior group estimates also indicated that drift rates differed across the 3 task conditions, such that No-Conflict Positive and Negative trials were respectively associated with more positive and negative drift rates than the other conditions (in both groups $P_{Positive > Negative}$ = 0.999, $P_{Positive > Conflict}$ > 0.999,

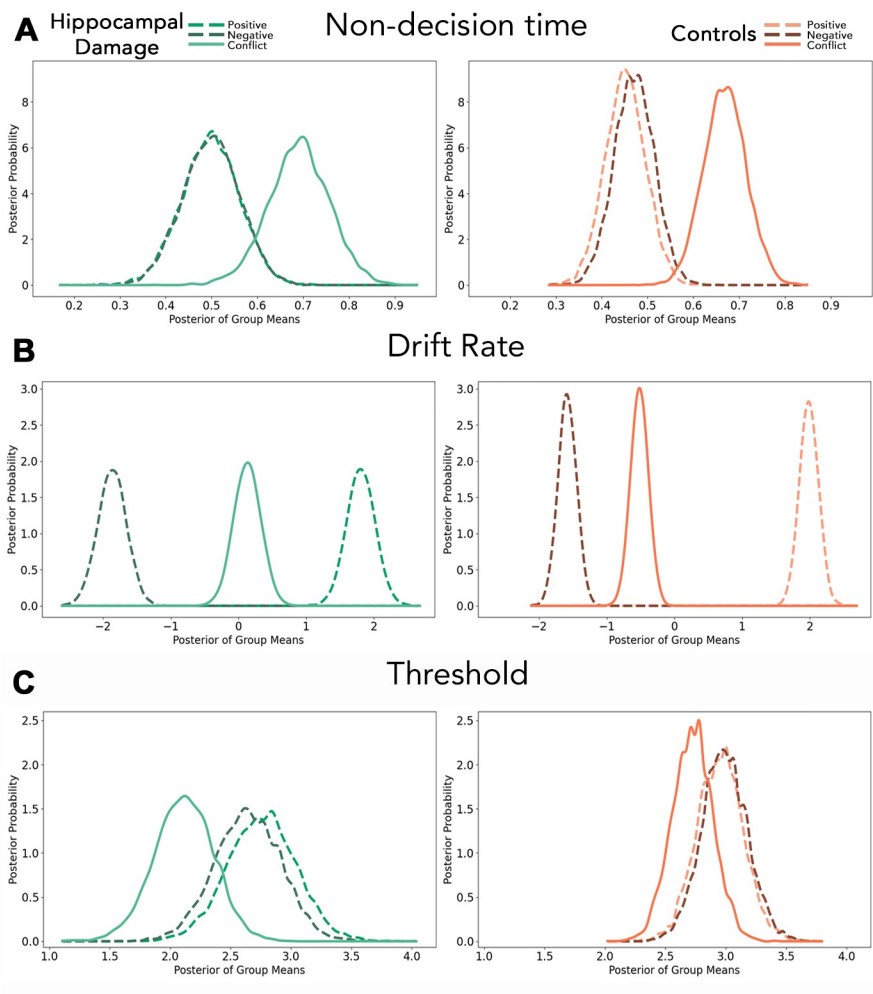

**Fig 3.** HDDM posterior distributions of means for (A) Non-decision time; (B) Drift rate; and (C) Threshold for hippocampal damage participants (green) and controls (red/orange). Data for hDDM analyses are available from https://doi.org/10.5683/SP3/C4GWZU. hDDM, hierarchical drift diffusion model.

$P_{Negative < Conflict} > 0.999$; Fig 3B). There was weak evidence that Conflict was associated with numerically smaller threshold values than either of the No-Conflict trials, with this difference being more likely for the comparison between No-Conflict Positive and Conflict trials in the hippocampal damage group (Hippocampal Damage: $P_{Positive > Conflict} = 0.956$, $P_{Negative > Conflict} = 0.927$; Controls: $P_{Positive > Conflict} = 0.915$, $P_{Negative > Conflict} = 0.856$; Fig 3C). Importantly, though, threshold values for No-Conflict conditions were similar in both groups (Hippocampal Damage: $P_{Positive > Negative} = 0.621$; Controls: $P_{Positive > Negative} = 0.441$). In aggregate, these findings are consistent with the notion that AAC decision-making, relative to No-Conflict decision-making, is characterized by increased recruitment of non-decision cognitive processes and slower evidence accumulation, and possibly with lower decision thresholds.

**Between-groups comparisons.** On both No-Conflict Positive and No-Conflict Negative trials, posterior group estimates indicated similar values for individuals with hippocampal damage and controls for non-decision time ($P_{Hippocampal Damage < Controls}$: No-Conflict Positive = 0.265, No-Conflict Negative = 0.350), drift rate ($P_{Hippocampal Damage < Controls}$: No-Conflict Positive = 0.766, No-Conflict Negative = 0.874), and threshold ($P_{Hippocampal Damage < Controls}$; No-Conflict Positive = 0.723, No-Conflict Negative = 0.858). However, there was very strong evidence for differences between the groups' parameter estimates on Conflict trials. Specifically, the hippocampal damage group drift rate was more positive relative to controls ($P_{Hippocampal Damage > Controls} = 0.995$; Fig 4A) and the hippocampal damage group exhibited lower decision thresholds compared to controls ($P_{Hippocampal Damage < Controls} = 0.982$; Fig 4B). We observed little evidence that the groups differed on non-decision time ($P_{Hippocampal Damage < Controls} = 0.391$) on Conflict trials. Lastly, we found strong evidence that starting biases were more positive for hippocampal damage participants than controls ($P_{Hippocampal Damage > Controls} = 0.992$; Fig 4C), suggesting a baseline approach propensity among the hippocampal damage group.

Taken together, these analyses suggest that individuals with hippocampal damage and controls did not differ markedly in their evidence accumulation processes during decision-making on No-Conflict trials. On Conflict trials, however, individuals with hippocampal damage lacked the rapid evidence accumulation towards avoidance seen in controls (controls' drift rate estimates were strongly negative, while hippocampal damage participants' estimates were close to 0), and they were willing to make decisions with less evidence than controls (i.e., lower threshold estimate). There was also a greater baseline bias towards approach decisions in individuals with hippocampal damage compared to controls (i.e., more positive bias estimate).

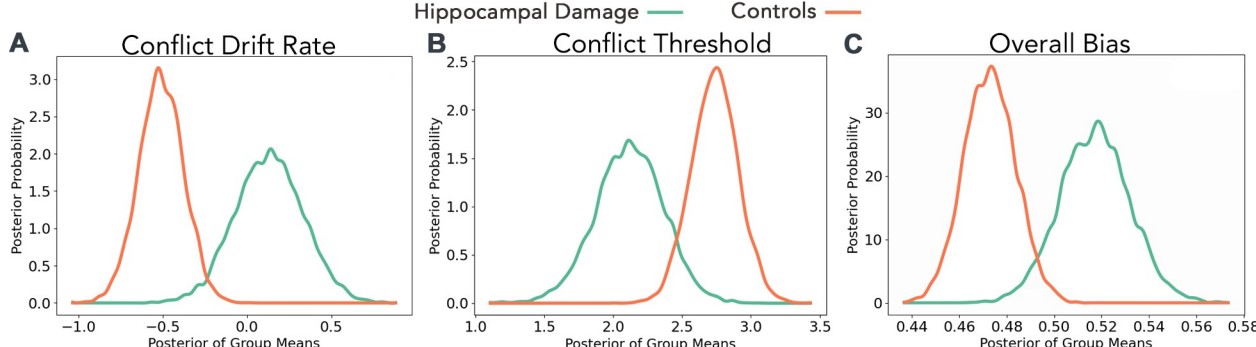

**Fig 4.** Examination of hDDM posterior distributions revealed that there was very strong evidence for differences between the hippocampal damage (green) and control (orange) groups for (A) Conflict drift rate ($P_{Hippocampal Damage > Controls} = 0.995$); (B) Conflict threshold ($P_{Hippocampal Damage < Controls} = 0.982$); and (C) overall starting bias ($P_{Hippocampal Damage > Controls} = 0.992$). Data for hDDM analyses are available from https://doi.org/10.5683/SP3/C4GWZU. hDDM, hierarchical drift diffusion model.

### Response conflict tasks

**Stroop task.** Participants were administered a computer-based version of the color Stroop task [24] in which they indicated the color of a rectangle (Control trials) or the lettering of words of color names presented on each trial via a key press. The color of the lettering and color name could either be congruent or incongruent. We analyzed accuracy on this task using a LMM with fixed effects of condition (Control versus Congruent versus Incongruent) and group, as well as the interactions between them, and random intercepts per participant. The formula for the selected model was: accuracy ~ incongruent_vs_control + incongruent_vs_congruent + group + group * incongruent_vs_control + group * incongruent_vs_congruent + (1 | participant). Table 5A summarizes the selected model's outputs, effect tests for multi-parameter predictors, and post hoc EMM comparisons (adjusted for multiple comparisons using Tukey's HSD).

As expected, a multi-parameter test revealed a significant main effect of condition ($p_{corr} <$ 0.001; Fig 5A). Post hoc comparisons indicated that this was driven by lower accuracy on Incongruent compared to Congruent trials ($p_{corr} < 0.001$) as well as to Control trials ($p_{corr} < 0.001$), but no difference in accuracy between Congruent and Control trials ($p_{corr} = 1.000$), suggesting that Incongruent trials produced greater response conflict than the other conditions. However, there was no significant group effect or group-by-condition interaction (all $p_{corr} = 1.000$), suggesting that the groups did not differ in their overall accuracy, nor in their ability to inhibit incorrect responses on Incongruent trials.

Response time data were analyzed with a LMM identical in structure to that described for accuracy using the following formula: RT ~ incongruent_vs_control + incongruent_vs_congruent + group + group * incongruent_vs_control + group * incongruent_vs_congruent + (1 | participant). Table 5B summarizes the selected model's outputs, effect tests for multi-parameter predictors, and post hoc EMM comparisons (adjusted for multiple comparisons using Tukey's HSD).

Here too, a multi-parameter test revealed the expected main effect of condition ($p_{corr} <$ 0.001; Fig 5B). Post hoc comparisons indicated that this was driven by significantly longer response times on Incongruent compared to Congruent trials ($p_{corr} = 0.014$) and compared to Control trials ($p_{corr} < 0.001$), but no difference between Congruent and Control trials ($p_{corr} =$ 1.000). This likely reflects the additional deliberation time needed to resolve the response conflict elicited by Incongruent trials compared to the other conditions. As with accuracy, we found no group differences, nor group-by-condition interactions (all $p_{corr} = 1.000$), suggesting that both groups responded at similar speeds across all conditions.

### Go/No-Go task

Participants were administered a Cued Go/No-Go task from the literature [25] in which they were presented with a rectangle (in vertical or horizontal orientation) on each trial and were required to either make a key press in response to a "Go" cue (rectangle turning green in color) or withhold from responding following a "No-Go" cue (rectangle turning blue in color). Importantly, a vertically oriented rectangle was associated with a 4:1 Go/No-Go trial ratio whereas the horizontal rectangle had a 1:4 Go/No-Go trial ratio. As we were interested in assessing response inhibition under response conflict, we analyzed the proportion of inhibition errors participants committed on No-Go trials (i.e., the proportion of these trials on which they incorrectly produced a response). To this end, we constructed a LMM with fixed effects of group and cue (Go versus No-Go), as well as the interactions between them, and random slopes per participant using the following formula: inhibition_errors ~ group * cue + (1 | participant). Table 6 summarizes the selected model's outputs.

**Table 5. Stroop task LMM results for (A) Accuracy; and (B) RT.** Since there were 3 trial types (Control, Congruent, Incongruent), condition was coded as a contrast between Incongruent and each of the other conditions (Control, Congruent). Significant predictors involving these contrasts were then explored further with a multi-parameter effects test to assess the significance of the main effect of condition. All post hoc EMM comparisons are adjusted for multiple comparisons using Tukey's HSD ($p_{HSD}$). To correct for multiple LMMs being conducted in this study (7 in total), all $p$-values have additionally been adjusted with a Bonferroni correction ($p_{corr}$). Significant Bonferroni corrected findings are highlighted in bold (* < 0.05; ** < 0.01, *** < 0.001).

| A. Accuracy | | | | | |
|---|---|---|---|---|---|
| *Model Summary* | | | | | |
| Predictor | $b$ | 95% CI | | $p$ | $p_{corr}$ |
| (Intercept) | 0.97 | 0.95–0.98 | | <0.001 | **<0.001**\*** |
| Control | 0.08 | 0.06–0.10 | | <0.001 | **<0.001**\*** |
| Congruent | 0.08 | 0.05–0.10 | | <0.001 | **<0.001**\*** |
| Group | −0.00 | −0.03–0.03 | | 0.782 | 1.000 |
| Control * Group | 0.02 | −0.02–0.06 | | 0.412 | 1.000 |
| Congruent * Group | 0.00 | −0.04–0.04 | | 0.913 | 1.000 |
| Random Effects | | | | | |
| $\sigma^2$ | 0.03 | | | | |
| $\tau_{00\ participant}$ | 0.00 | | | | |
| Intraclass Correlation Coefficient | 0.02 | | | | |
| N | 29 | | | | |
| Observations | 2,436 | | | | |
| Marginal $R^2$ / Conditional $R^2$ | 0.040 / 0.062 | | | | |
| *Multi-Parameter Effect Tests* | | | | | |
| Effect | Predictors | $\Delta_{AIC}$ | $\Delta_{BIC}$ | $\chi^2$(df) | $p$ | $p_{corr}$ |
| Condition main effect | Control, Congruent | 65.80 | 54.20 | 69.79 (2) | <0.001 | **<0.001**\*** |
| *Post hoc EMM Comparisons* | | | | | |
| Contrast | Estimate (SE) | $t$ (df) | $p_{corr}$ | $p_{corr}$ | Cohen's $d$ | 95% CI |
| Incongruent–Control | −0.08 (0.01) | −7.43 (2,405) | <0.001 | **<0.001**\*** | 0.45 | [0.33, 0.57] |
| Incongruent–Congruent | −0.08 (0.01) | −7.04 (2,405) | <0.001 | **<0.001**\*** | 0.43 | [0.31, 0.55] |
| Control–Congruent | 0.00 (0.01) | 0.38 (2,405) | 0.981 | 1.000 | 0.02 | [−0.10, 0.15] |
| B. Response Time | | | | | |
| *Model Summary* | | | | | |
| Predictor | $b$ | 95% CI | | $p$ | $p_{corr}$ |
| (Intercept) | 1,815.21 | 1,416.59–2,213.84 | | <0.001 | **<0.001**\*** |
| Control | −480.11 | −703.26 – −256.96 | | <0.001 | **<0.001**\*** |
| Congruent | −408.04 | −631.19 – −184.89 | | <0.001 | **0.002**\** |
| Group | −526.20 | −1,323.45–271.04 | | 0.196 | 1.000 |
| Control * Group | 108.86 | −337.43–555.16 | | 0.632 | 1.000 |
| Congruent * Group | 135.11 | −311.19–581.41 | | 0.553 | 1.000 |
| Random Effects | | | | | |
| $\sigma^2$ | 34,676,618.87 | | | | |
| $\tau_{00\ participant}$ | 745,110.23 | | | | |
| Intraclass Correlation Coefficient | 0.18 | | | | |
| N | 29 | | | | |
| Observations | 2,436 | | | | |
| Marginal $R^2$ / Conditional $R^2$ | 0.025 / 0.197 | | | | |
| *Multi-Parameter Effect Tests* | | | | | |
| Effect | Predictors | $\Delta_{AIC}$ | $\Delta_{BIC}$ | $\chi^2$(df) | $p$ | $p_{corr}$ |
| Condition main effect | Control, Congruent | 17 | 5 | 20.84 (2) | <0.001 | **<0.001**\*** |
| *Post hoc EMM Comparisons* | | | | | |
| Contrast | Estimate (SE) | $t$ (df) | $p_{HSD}$ | $p_{corr}$ | Cohen's $d$ | 95% CI |

*(Continued)*

**Table 5.** (Continued)

| Incongruent–Control | 480.10 (114) | 4.22 (2,403) | <0.001 | <**0.001**\*** | 0.26 | [0.14, 0.38] |
| Incongruent–Congruent | 408 (114) | 3.59 (2,403) | 0.002 | **0.014**\* | 0.22 | [0.10, 0.34] |
| Control–Congruent | −72.10 (115) | −0.62 (2,403) | 0.924 | 1.000 | 0.04 | [−0.16, 0.08] |

EMM, estimated marginal mean; HSD, honestly significant difference; LMM, linear mixed model.

As expected, No-Go trials with Go cues were associated with numerically higher error rates than No-Go trials with No-Go cues (Fig 5C), suggesting that the former elicited significantly greater difficulties with response inhibition than the latter. This effect, however, did not survive Bonferroni correction ($p = 0.016$; $p_{corr} = 0.112$). There were no significant group or group-by-cue effects (both $p_{corr} \geq 0.882$), suggesting that the groups did not differ in their ability to inhibit responses on No-Go trials, regardless of cue type.

## Discussion

We compared the behavior of individuals with HPC lesions to that of healthy controls on an AAC paradigm and employed computational modeling to elucidate the underlying latent cognitive processes. Both groups approached and avoided No-Conflict positive and negative trials at similar rates, and exhibited longer response times on Conflict trials, suggesting the successful establishment of motivational conflict. Crucially, however, the individuals with hippocampal damage approached significantly more often than control participants on these trials associated with high AAC. There was limited evidence for differences in HDDM parameters across groups under No-Conflict conditions, with strong evidence for group differences on Conflict trials only. Specifically, there was strong evidence that relative to controls, individuals with hippocampal damage exhibited lower decision thresholds during AAC while controls exhibited faster evidence accumulation towards avoidance than hippocampal damage participants. The hippocampal damage group also demonstrated a stronger general positive bias, indicating a greater overall predilection towards approach decisions, regardless of condition. Taken together, our findings provide strong evidence that structural damage to the HPC potentiates approach behavior under AAC and that this is driven by an increased baseline propensity to approach, a willingness to initiate decisions with less accumulated evidence than controls, and a slower-than-typical drift toward avoidance.

Although we sought to minimize mnemonic demands, participants were nevertheless required to learn and recall stimulus-valence associations in our AAC paradigm. Our findings

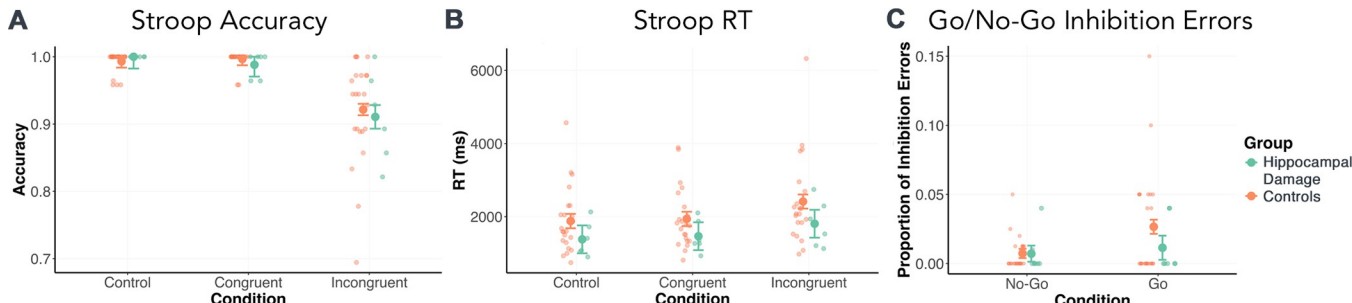

**Fig 5.** No significant group differences were observed for (A) Stroop task accuracy ($p_{corr} = 1.000$); (B) Stroop task response times ($p_{corr} = 1.000$); or (C) Go/No-Go Task proportion of inhibition errors ($p_{corr} = 1.000$) (individual data points with EMMs ±SE). Underlying data for these figures and associated analyses are available from https://doi.org/10.5683/SP3/C4GWZU. EMM, estimated marginal mean.

**Table 6. Go/No-Go task inhibition errors LMM results.** To correct for multiple LMMs being conducted in this study (7 in total), all $p$-values have additionally been adjusted with a Bonferroni correction ($p_{corr}$). Significant Bonferroni corrected findings are highlighted in bold (* < 0.05; ** < 0.01, *** < 0.001).

| Model Summary | | | | |
|---|---|---|---|---|
| Predictor | $B$ | 95% CI | $p$ | $p_{corr}$ |
| (Intercept) | 0.01 | 0.01–0.02 | <0.001 | **0.005**** |
| Group | −0.01 | −0.02–0.01 | 0.278 | 1.000 |
| Cue | 0.01 | 0.00–0.02 | 0.016 | 0.112 |
| Group * Cue | −0.01 | −0.03–0.00 | 0.126 | 0.882 |
| Random Effects | | | | |
| $\sigma^2$ | 0.01 | | | |
| $\tau_{00\ participant}$ | 0.00 | | | |
| Intraclass Correlation Coefficient | 0.01 | | | |
| N | 30 | | | |
| Observations | 3,343 | | | |
| Marginal $R^2$ / Conditional $R^2$ | 0.005 / 0.018 | | | |

LMM, linear mixed model.

cannot, however, be attributed to group differences in memory ability, a critical point given the role of the HPC in mnemonic processing [26,27]. Both groups showed similar accuracy in valence judgments of the individual stimulus images at the end of the learning phases and crucially, similarly approached No-Conflict Positive pairs and avoided No-Conflict Negative pairs nearly 100% of the time during the decision phase. This indicates that individuals with hippocampal damage and controls were able to appropriately recall the valences of individual images to inform their decisions and that the hippocampal damage participants' more frequent approach behavior on Conflict trials cannot be explained by poorer stimulus valence memory (e.g., for negative stimuli).

Given that the individuals with hippocampal damage in our study were selected on the basis of relatively circumscribed volume loss to the HPC, the observed AAC behavioral differences can be attributed to structural alterations to this area. Our causal findings dovetail with correlative evidence from human neuroimaging studies and lesion findings from nonhuman primates highlighting a role for the HPC in arbitrating AAC decisions [10,12,28]. Moreover, they are not inconsistent with previous findings that restricted lesioning of the rodent vHPC potentiates approach behavior [7], as the hippocampal damage participants' volume loss was numerically slightly greater within the anterior portion of the HPC compared to the posterior portion. The current data also add considerably to existing neuropsychological studies on AAC conflict in humans, which have involved behavioral tasks with a spatial navigation component and/or a single patient participant with circumscribed bilateral HPC damage [11,13].

Notably, the present study provides novel insight into the role of the HPC in AAC by demonstrating that it is critically involved in the evidence accumulation processes that underlie AAC decision-making. One open question is why a disruption to HPC-dependent evidence accumulation leads to a potentiation of approach rather than avoidance under conditions of AAC. The former is suggestive of an inhibitory role of the HPC in resolving AAC, which is consistent with interpretations of previous rodent work, wherein the vHPC has been shown to be involved in anxiogenic behaviors and cost-benefit evaluations, with vHPC ablation producing disinhibited behavior [29–38]. One theoretical model suggests that the HPC acts as a comparator between current and predicted event states (e.g., outcome), and that conflict is detected when there is an incongruency between the two (e.g., reward versus punishment), resulting in preferential strengthening of representations of negative possible outcomes and

subsequently in behavioral inhibition [39]. Broadly speaking, our finding of group differences in drift rates is consistent with this idea: whereas normal HPC functioning under conflict is associated with rapid integration of information to inform an avoidant decision, structural damage to the HPC appears to blunt this process, resulting in more neutral drift rates. Indeed, the strongly negative drift rates observed in controls may reflect the predominant retrieval of evidence associated with undesirable outcomes. Likewise, the finding of reduced decision thresholds under conflict within the hippocampal damage group is in keeping with a behavioral inhibition model. While our hDDM analyses offered some evidence that AAC may generally be associated with lower decision thresholds, this was especially true within the hippocampal damage group. That the individuals with hippocampal damage were initiating responses on Conflict trials with less evidence than controls may reflect a disruption of hippocampally mediated comparison of possible outcome states.

Although not explicitly predicted, the finding of a greater general approach bias among individuals with hippocampal damage is not incompatible with a behavioral inhibition viewpoint and suggests that the HPC may exert some inhibitory influence even prior to the initiation of evidence accumulation. Although speculative, it may be that, even at baseline, the HPC holds representations of possible aversive outcomes associated with goal-relevant stimuli. Damage to the HPC may disrupt these representations and produce a more reward-seeking and less cautious disposition. It may also be that the disruption of these representations hampers the very detection of motivational conflict, as their inclusion in the comparison of possible outcome states is theoretically integral to this process.

It is important to note that, while our findings are broadly consistent with a behavioral inhibition model, it is also clear that the HPC's role in AAC processing is not fully captured by this viewpoint. For instance, ventral CA1 inactivation has been reported to increase avoidance, with CA3 inactivation increasing approach under AAC conditions [8]. Likewise, increased aHPC fMRI activity has been observed for approach behavior, which is somewhat unexpected if this region is involved in the wholesale inhibition of approach responses [12]. Furthermore, human electrophysiological work has demonstrated that firing rates in the HPC following rewarding and punishing outcomes during an AAC task predict subsequent approach and avoid decisions [40]. Finally, the HPC has also been implicated in evidence sampling to support choice behavior in the face of competing positive outcomes [41]. Taken together, these findings suggest that the involvement of the v/aHPC in AAC processing is likely more complex than the exertion of inhibitory control, and that subregions within the aHPC, along with their distinct extra-hippocampal connectivity [9], contribute differentially to AAC resolution in ways that remain to be elucidated.

Considering prior work demonstrating that the HPC is preferentially involved in the processing of spatial and contextual stimuli, and that perirhinal cortex (PRC) is preferentially involved in the processing of objects [42–44], it is surprising that the group differences on the AAC paradigm were not specific to the Scene task, particularly considering the absence of significant PRC volume group differences (Table 1). Indeed, recent human neuroimaging and rodent optogenetic studies have found that PRC, rather than the v/aHPC, is predominantly involved in the resolution of AAC associated with discrete objects [18,19]. Although the reason for this contradiction is unknown, it may relate to findings that have highlighted that the relationship between volume loss and impaired memory in hippocampal amnesia is mediated by abnormalities in functional connectivity [45]. In a similar fashion, network abnormalities, in addition to HPC volume loss, may be contributing to the observed patterns of AAC behavior. For instance, the v/aHPC and d/pHPC and their subregions possess different patterns of connectivity [46,47] and thus, it is conceivable that the impact of HPC damage on AAC processing is dependent on the location and extent of cell loss, and the associated disruption to wider

anatomical and functional networks. Due to our selection criteria for participants with hippocampal damage, we did not have sufficient numbers to analyze network functioning in this study and future research will need to address this issue by potentially incorporating a less mnemonically demanding AAC task to allow the inclusion of a greater number of hippocampal damage participants.

One important question moving forwards is to what extent the present findings can be generalized to other AAC behavioral paradigms. For instance, while the current study used a secondary reinforcer (i.e., game points) and stimuli for which the incentive values had to be learned, many nonhuman animal AAC tasks involve innate rewards or threats, including ethological tests of anxiety or behavioral tasks involving predator stimuli. Since prior experience of innately conflicting stimuli may be very limited, it is possible that impaired conflict detection following HPC dysfunction may be the primary contributor towards increased approach behavior in these latter paradigms [10,33,48] compared to the disrupted retrieval of outcome evidence for learned AAC scenarios. With respect to human AAC behavior, a broad range of tasks have been used including human adaptations of rodent ethological tests of anxiety such as the open field test [11,49] and elevated plus maze [50], operant conflict tasks in which a lever/button press can be associated with both reward and punishment [51–53], and gambling-like tasks [28], to name a few. While there are fundamental characteristics that are shared between these tasks (e.g., the possibility of receiving reward and punishment), it is also evident that there are clear differences such as the nature of conflict elicited, and the type and schedule of reinforcers used. Whether the HPC contributes to AAC processing in a similar manner across these paradigms, therefore, remains to be investigated. Indeed, extending this line of thought more broadly, to what extent the current HPC findings are relevant to mental health disorders, in particular anxiety, is also unclear. The relationship between AAC and clinical anxiety is underexplored and likely complex, and limitations pervade the behavioral paradigms that have typically been used to study this disorder in preclinical and clinical models [54–56].

Lastly, we also administered Stroop and Go/No-Go tasks to determine whether group differences in AAC behavior extended to response conflict tasks. Across a number of measures, no significant group differences were found on either task, which contrasts with previous work that has reported greater HPC activity during high response conflict Stroop [22,57], Flanker, and Garner filter interference [58] trials, as well as higher response error rates on high response conflict Stroop trials, which were correlated with right HPC volume loss in medial temporal lobe epilepsy (MTLE) patients [23]. Although accounting for this discrepancy is beyond the aims of this study, it is worth noting that greater HPC activity during response conflict may reflect, at least in part, greater incidental memory encoding [59] (although see also [60]).

In summary, our findings indicate that structural damage to the HPC in humans results in changes in AAC decision-making behavior and the associated underlying evidence accumulation processes. By assessing multiple participants with relatively circumscribed HPC lesions and leveraging computational modeling techniques, our study provides novel, robust causal evidence of a role for the HPC in arbitrating AAC behavior.

## Materials and methods

### Participants

Ten participants were recruited as part of the hippocampal damage group. One of these participants has medial temporal lobe epilepsy (MTLE) with sclerosis to the HPC (Table 1) and mild amnesia. The other 9 individuals had previously participated in a larger neuroimaging study [45] following a rare form of autoimmune limbic encephalitis (aLE) that is associated with an

increase in antibodies against the voltage-gated potassium channel (VGKC) complex [61]. These aLE patients were selected for this study on the basis of their relatively circumscribed focal HPC atrophy (Table 1) and their mild amnestic profile as captured by standard neuropsychological tests (Table 2). Two of these patients, however, struggled with the mnemonic demands of the experimental tasks and were therefore excluded. Of the remaining 7 aLE patients, one patient's data set was excluded from analyses for the Scene AAC task due to a failure to learn the stimulus valences and another was removed from the Stroop task due to a misunderstanding of task instructions—the data for these patients were otherwise retained for all other analyses.

Twenty-six adults were recruited as healthy control participants (2 to 3 age-matched participants per participant with hippocampal damage) and were required to have normal or corrected-to-normal vision and hearing, and no previous or current neurological condition or traumatic brain injury. One individual had recently recovered from a stroke and was, therefore, ineligible to be included. One control data set was excluded from each of the Scene and Object AAC tasks due to a failure to learn stimulus valences, and one data set was removed from each of the Stroop and the Go/No-Go tasks due to a misunderstanding of task instructions—all other data sets from these control participants were otherwise included.

Our final sample, therefore, comprised 8 individuals with hippocampal damage (7 male, 1 female; Age: $M = 63.90$, $SD = 8.32$; IQ: $M = 109.00$, $SD = 14.90$) and 25 control participants (14 male, 11 female; Age: $M = 68.5$, $SD = 9.24$; IQ: $M = 107.84$, $SD = 11.10$). The groups did not differ significantly in either age ($t(13.02) = 1.34$, $p = 0.204$) or IQ as measured by the Wechsler Abbreviated Scale of Intelligence, Second Edition (WASI-II) [62] ($t(9.63) = 0.20$, $p = 0.844$).

Experimental task data were collected from the individuals with hippocampal damage over the course of a single session, either in their homes or at the John Radcliffe Hospital in Oxford, United Kingdom. Control participants underwent 2 testing sessions on separate days to collect experimental and background neuropsychological task data. Due to COVID-19–related restrictions, 13 controls completed one or both testing sessions virtually, during a synchronous Zoom session (https://zoom.us). The remaining 12 participants were tested in person at the University of Toronto Scarborough campus. All participants gave informed written consent prior to participation and received monetary compensation plus travel/parking expenses, if applicable. This study received ethical approval from the University of Toronto Research Ethics Board (#26827) and the South Central Oxford Research Ethics Committee (#08/H0606/133) and was conducted in accordance with the ethical principles expressed in the Declaration of Helsinki.

## Background neuropsychology

The individuals with hippocampal damage had previously undergone a standard neuropsychological battery (aLE patient data from [45]). A comparable battery was devised for the control group, which included subtests selected from the following neuropsychology tests: the Montreal Cognitive Assessment (MoCA) [63], the Wechsler Memory Scale, Fourth Edition (WMS-IV) [65], the Doors and People test battery (D and P) [66], the Rey Complex Figure Task (RCFT) [67], the WASI-II [62], and the Visual Object Spatial Perception Battery (VOSP) [68].

Due to technical difficulties and 2 participants not returning for a second session, not all control participants completed all neuropsychology tests. Considering performance across neuropsychology subtests, all controls retained in the sample were deemed neurologically healthy, performing in aggregate in the "Average" to "Superior" range across all subtests (Table 2).

## Experimental procedure

Participants completed 2 experimental AAC tasks (Object and Scene) and 2 response conflict tasks (Stroop and Go/No-Go) within the same testing session, the order of which was counter-balanced across participants. Control participants additionally completed the MoCA before the experimental tasks in the first session, and the remaining neuropsychological tests after the experimental tasks in the remainder of the first session and in the second session.

## In-person versus online testing sessions

Data collection was interrupted by the COVID-19 pandemic, which precluded in-person testing. The testing protocol was therefore adapted to allow for synchronous remote experimental sessions. For in-person testing, we programmed AAC tasks using E-Studio version 2.0.10.252 from the E-Prime 2 Professional Suite (https://pstnet.com) while the Stroop and Go/No-Go tasks were programmed and administered in Inquisit version 5.0.12.0 (https://www.millisecond.com). Tasks were administered on a 12' Lenovo ThinkPad X240 laptop (2.2 GHz Intel Core i7-4600U CPU processor; 8GB RAM; 1,366 × 768-pixel monitor) and occupied a 1,024 × 768-pixel window on-screen.

For synchronous remote data collection, participants connected with the experimenter via Zoom videoconferencing software for the duration of both sessions. AAC, Stroop, and Go/No-Go tasks were re-created using PsychoPy Experiment Builder v2020.2.8 [69] and administered in participants' browsers through Pavlovia (https://pavlovia.org).

Neuropsychology tasks were adapted to replicate in-person testing conditions as faithfully as possible. Images normally presented in stimulus books were scanned and presented using Microsoft PowerPoint (https://www.microsoft.com) via screen-sharing. If participants were required to draw as part of a task, the image was held up to the camera and screen-captured by the experimenter, and then mailed or scanned and emailed to the experimenter.

## Approach-avoidance conflict tasks

AAC processing was assessed using 2 different tasks (Object and Scene) to determine whether HPC-mediated AAC processing effects are specific to certain classes of complex everyday stimuli, based on prior work showing regional specificity in visual stimulus class processing [42–44], which may extend to AAC processing [18,19]. These tasks were simplified versions of previously used AAC paradigms [12,16,18] and were identical in structure—consisting of an initial learning phase and a subsequent decision phase—but differed in the stimuli presented (Fig 1A). Both tasks consisted of 3 learning blocks of 40 trials each, and 3 test blocks of 36 trials each. On the Object task, the stimulus images depicted unfamiliar computer-generated objects (in-person image dimensions = 384 × 288 pixels; remote dimensions = 50% × 37% participants' display height, DH). On the Scene task, the stimuli consisted of photographs of real-world unfamiliar scenes that were easily recognizable, but otherwise nondescript (i.e., no famous landmarks or monuments, or people present; in-person dimensions = 653 × 357 pixels; remote dimensions = 85% × 46% DH). Participants were given instructions verbally and via written text using Microsoft PowerPoint prior to each task, as well as a text-based refresher of the instructions prior to each block of trials (on-screen text: black Courier New font, size 18, over white background).

On all blocks across both tasks, participants were instructed to earn as many game points as possible. In the learning phases, participants were required to learn the valences associated with stimulus images (positive or negative) through trial-and-error. To minimize mnemonic demands on these tasks, participants learned only 4 stimulus-valence associations per task (i.e., 2 per valence; Fig 1A). On each learning phase trial (Fig 1B), a single stimulus image was

presented in the center of the screen, enclosed in a black line border (live testing: border size = 768 × 576 pixels, line size = 10 pixels; remote testing: 100% × 75% DH, line size = 10 pixels). Participants chose whether to approach or to avoid the image by pressing the "1" or "2" keys, respectively. Image-valence associations were arbitrary and predetermined. Approaching a positive image resulted in a gain of 100 game points, while avoiding it resulted in no gain in points. Approaching a negative image resulted in a loss of 100 points, while avoiding it resulted in no loss of points. Therefore, to maximize their score, participants needed to approach positive images, and avoid negative ones. Stimulus images remained on-screen until participants responded, after which (latency = 500 ms) participants were shown text-based feedback on the outcome of their response (duration = 1,500 ms), which included the number of points lost/gained, a running sum of their score, and, if they avoided, a message indicating whether they had correctly avoided a negative image ("Good") or had mistakenly avoided a positive image ("Miss"). A 1,000 ms inter-stimulus interval consisting of a white screen then ensued. The 4 stimulus images were presented in a random order in mini blocks of 4, for a total of 30 times each (10 per block of 40 trials) in the learning phase.

In the decision phase (Fig 1C), stimuli from the learning phase were combined into 3 possible pairs: No-Conflict Positive (2 positive images) (12 trials per block), No-Conflict Negative (2 negative images) (12 trials per block), or Conflict (1 image of each valence) (12 trials per block). Trials consisted of the concurrent presentation of 2 images, enclosed in a black line border (dimensions identical to those described above). Image pairs remained onscreen until participants chose to approach/avoid them via a key press, after which a 1,000 ms ISI (white screen) occurred. Participants were told that approaching a No-Conflict Positive pair or a No-Conflict Negative pair would result in a gain or loss of 100 points, respectively, and that avoiding any pair would result in no change in score. As such, the appropriate responses were to approach No-Conflict Positive pairs and to avoid No-Conflict Negative Pairs. Participants were told that approaching on Conflict trials was equally likely to result in either a gain or loss of 100 points. Participants were therefore instructed to decide whether to approach, weighing the risk of losing points with the possibility of gaining points. They were also told that to maximize their score, they would have to approach on at least some Conflict trials. However, participants' scores were not tracked. To prevent learning and consistent with our previous work [12,16,18], no feedback was given following their responses during the decision phase.

### Response conflict tasks

**Stroop task.** Participants completed a computer-based version of the Stroop task [24]. On every trial, either a word spelling out the name of a color (live and remote testing: font size = 7% DH) or a colored rectangle (live and remote testing: dimensions = 20% × 10% DH) appeared on the screen. The word could appear in either green, red, blue, or black lettering. Participants were instructed to indicate the color of the lettering (while ignoring the name of the color the word spelled) or the rectangle by pressing a corresponding key ("f," "d," "j," and "k," respectively). There were 3 conditions. In the Congruent condition, the color name and the color of the lettering were the same (e.g., "green" spelled in green letters). In the Incongruent condition, the color name and the color of the lettering were different (e.g., "black" spelled in red letters). Trials with rectangles were the Control condition. Participants completed 4 trials in each of the 3 conditions, over 7 repetitions, for a total of 28 trials per condition, and 84 trials overall. If the participant responded correctly, the next trial was immediately initiated (ITI = 200 ms). If they responded incorrectly, a red "X" flashed in the center of the screen (duration = 400 ms). Participants were given as much time as they needed to respond but were instructed to work as quickly as they could, while making as few mistakes as possible.

**Cued Go/No-Go task.**   We implemented a Cued Go/No-Go task from the literature [25]. On every trial, participants viewed a fixation cross (duration = 500 ms; live and remote dimensions = 10% × 10% DH), followed by a white rectangle (i.e., the cue; delay = 300 ms). The rectangle appeared in either a horizontal or vertical orientation (live and remote dimensions = 30% × 10% DH or vice-versa, depending on orientation). After a brief pause (SOA = [100, 500 ms]), the rectangle turned either green or blue. If the rectangle turned green, participants were instructed to press the spacebar as quickly as possible. If the rectangle turned blue, they were instructed to do nothing and wait for the next trial (ITI = 400 ms). Trials ended after 700 ms, if participants did not respond. The orientation of the cue rectangle related to the likelihood of a Go or No-Go trial: vertical cues had a 4:1 Go/No-Go trial ratio while horizontal cues had a 1:4 Go/No-Go trial ratio. Participants completed 250 trials.

## Statistical analyses

**Linear mixed models.**   We analyzed individual trial choice and response time data for the AAC, Stroop, and Go/No-Go tasks using LMMs, as implemented by the lme4 package [70] in R 4.0.4 (R Core Team, 2021) (to be precise, generalized LMMs were used for choice data given their categorical nature, e.g., approach versus avoid, but will be collectively referred to as LMMs for simplicity). Besides the consideration of random effects, the use of LMMs allowed us to account for the unbalanced nature of our groups, potential inequalities in variance between groups, and the fact that some participants' data were excluded for one of the AAC tasks. LMMs were constructed iteratively, such that the greatest number of desired random effects were modeled, while achieving convergence and appropriate fit. We constructed models based on the protocol described in [71], evaluating the most complex models first. When a model converged and showed appropriate fit, it was compared both to a purely fixed effects model and to a simplified nested mixed effects model with a likelihood ratio test to determine whether the inclusion of random effects significantly improved model fit. In cases where multiple candidate models converged and showed appropriate fit, they were compared to one another with a likelihood ratio test, and the model with the best fit was selected. In cases where model fit did not differ significantly, the model of highest theoretical interest (i.e., with the highest number of random effects) was chosen.

Since all our predictors were categorical and we modeled the interactions between them, a deviation coding scheme was adopted as this facilitates the interpretation of main effects [72]. In models of AAC decision phase and Stroop task data, the predictors for trial condition each comprised 3 levels (No-Conflict Positive, No-Conflict Negative, Conflict; Congruent, Control, Incongruent) and therefore were coded using 2 dummy variables. In the AAC decision phase data, these represented contrasts between the Conflict condition and one of the No-Conflict conditions. In the Stroop task, these represented contrast between the Incongruent condition and the Congruent and Control conditions. Because the main and interaction effects involving condition in these analyses were represented by multiple model terms, we evaluated significant model predictors involving condition by performing multi-parameter tests, wherein, using a likelihood ratio test, the selected model was compared to an identical model with the terms for the relevant effects removed. If the specified model showed better fit than its counterpart with the relevant effect terms removed, the main or interaction effect was considered significant. Significant main effects and interactions were probed via post hoc comparisons between relevant EMMs using the emmeans package [73] and correcting $p$-values for multiple comparisons using Tukey's HSD. Lastly, since multiple LMMs were conducted across multiple behavioral tasks and response measures (7 LMMs in total), we accounted for the increased probability of Type I errors occurring by additionally adjusting all $p$-values using a Bonferroni correction

(i.e., $p \times 7$). Both uncorrected and corrected $p$-values are reported in the statistical tables (Tables 3–6).

**Hierarchical drift diffusion modeling.**   In addition to analyzing rates of approach responding, we were interested in whether the underlying decision-making processes differed across groups. To this end, we employed the hDDM. DDM approaches are a type of sequential sampling techniques, which have long been used to model two-choice behavior and response times [74]. These models assume that decision-making occurs by means of accumulation of noisy information (i.e., evidence) about the stimulus. In DDMs, evidence is continuously evaluated while it is collected, until sufficient information has been gathered to cross a decision threshold. DDMs produce parameter estimates for decision thresholds ($a$), the rate at which evidence toward either decision threshold is accumulated (i.e., drift rate; $v$), and the distance between the information gathering "starting point" and either threshold (i.e., bias; $z$). Additionally, these models provide estimates of non-decision time ($t$), or time between the presentation of the stimulus and the initiation of evidence accumulation. HDDM methods hold a significant advantage over traditional DDMs, in that they handle nested data structures more effectively. They also use Bayesian inference to produce full posterior distributions of parameter estimates, providing both the most likely value for a given estimate (i.e., the distribution's mode) and an indication of the relative certainty of the estimate (i.e., distribution's spread).

We used the hDDM software package in Python v3.8.10 (Python Software Foundation, 2021), which allows flexible construction of hDDM models, and which uses the PyMC package [75] to generate parameter distributions [21]. We generated a single model incorporating both groups and collapsing across Object and Scene tasks since we did not observe a significant effect of stimulus type in our LMM analyses of choice and response time data (see Results). Model selection was based on an iterative process, wherein the most complex models were evaluated first using several methods to assess model fit and convergence. These included visual inspection of parameter estimates posterior distribution and trace plots, examining whether parameter Gelman–Rubin $\hat{R}$ values fell below a specified cut-off of 1.01 (with $\hat{R} = 1$ representing perfect convergence) and comparing each parameter's Monte Carlo (MC) error statistic, to its posterior distribution's standard deviation, with MC error values less than 1% of the posterior considered to show poor convergence. For each specified model, we generated 55,000 samples from posterior distributions, of which 5,000 were discarded. Of the remaining 50,000 samples, we saved one of every five, yielding a net sampling trace length of 10,000 samples. The most complex and theoretically desirable model estimated separate parameters per trial condition for each participant. If a model did not achieve convergence, it was simplified (i.e., by assigning parameters with poor convergence indices to be estimated only at the group level) and re-evaluated. The selected model was compared to alternative models (i.e., with fewer parameters estimated per participant and per condition) to determine whether the inclusion of additional parameter estimates improved model fit. We assessed relative fit across models by comparing their DIC values. DIC is a Bayesian measure of model fit and is defined as a classical estimate of fit plus twice the effective number of parameters [76]. Lower DIC values indicate better fit, and models whose values differ by 10 or more are considered to show significantly different fit. To test hypotheses about differences in parameter estimates within and between groups, we examined the overlap between their posterior distributions to determine the probability, denoted by P, that a value drawn from either distribution was greater than the other. To illustrate, a hypothesis test of $P_{Positive} > P_{Negative} = 0.493$ indicates that a randomly drawn value from the posterior distribution for No-Conflict Positive trials has a 49.3% probability of being greater than a value drawn from the posterior distribution for the No-Conflict Negative condition. In other words, the estimated distributions are sufficiently

overlapping as to provide little evidence for a difference between No-Conflict Positive and Negative trials.

## Acknowledgments

We thank all participants for their time, and Yi Yang Teoh for help with the hDDM analyses.

## Author Contributions

**Conceptualization:** Rutsuko Ito, Andy C. H. Lee.

**Data curation:** Willem Le Duc.

**Formal analysis:** Willem Le Duc, Cendri Hutcherson.

**Funding acquisition:** Rutsuko Ito, Andy C. H. Lee.

**Investigation:** Willem Le Duc, Christopher R. Butler, Georgios P. D. Argyropoulos, Sonja Chu, Andy C. H. Lee.

**Methodology:** Sonja Chu, Andy C. H. Lee.

**Project administration:** Christopher R. Butler, Georgios P. D. Argyropoulos, Andy C. H. Lee.

**Resources:** Christopher R. Butler, Georgios P. D. Argyropoulos.

**Supervision:** Christopher R. Butler, Anthony C. Ruocco, Andy C. H. Lee.

**Visualization:** Willem Le Duc.

**Writing – original draft:** Willem Le Duc.

**Writing – review & editing:** Christopher R. Butler, Georgios P. D. Argyropoulos, Cendri Hutcherson, Anthony C. Ruocco, Rutsuko Ito, Andy C. H. Lee.

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
