## [Editor Report · Decision Letter 0]

13 Jun 2024

Dear Dr Lee, 

Thank you for submitting your manuscript entitled "Hippocampal Damage Disrupts the Latent Decision-Making Processes Underlying Approach-Avoidance Conflict Processing" for consideration as a Research Article by PLOS Biology.

Your manuscript has now been evaluated by the PLOS Biology editorial staff as well as by an academic editor with relevant expertise and I am writing to let you know that we would like to send your submission out for external peer review.

Once your full submission is complete, your paper will undergo a series of checks in preparation for peer review. After your manuscript has passed the checks it will be sent out for review. To provide the metadata for your submission, please Login to Editorial Manager (https://www.editorialmanager.com/pbiology) within two working days, i.e. by Jun 15 2024 11:59PM.

Kind regards,

Christian

Christian Schnell, PhD

Senior Editor

PLOS Biology

cschnell@plos.org

---

## [Decision Letter · Decision Letter 1]

31 Jul 2024

Dear Dr Lee,

Thank you for your patience while your manuscript "Hippocampal Damage Disrupts the Latent Decision-Making Processes Underlying Approach-Avoidance Conflict Processing" was peer-reviewed at PLOS Biology. It has now been evaluated by the PLOS Biology editors, an Academic Editor with relevant expertise, and by several independent reviewers. 

In light of the reviews, which you will find at the end of this email, we would like to invite you to revise the work to thoroughly address the reviewers' reports.

As you will see below, both reviewers are quite supportive of your study but have a few concerns about the statistical analyses and about the integration of the findings into the published literature. We would encourage you to take special care in addressing Reviewer 1's comment #1 and Reviewer 2's comments #3 and #5.

Given the extent of revision needed, we cannot make a decision about publication until we have seen the revised manuscript and your response to the reviewers' comments. Your revised manuscript is likely to be sent for further evaluation by all or a subset of the reviewers.

**IMPORTANT - SUBMITTING YOUR REVISION**

*Re-submission Checklist*

*Published Peer Review*

*PLOS Data Policy*

*Blot and Gel Data Policy*

Sincerely,

Christian

Christian Schnell, PhD

Senior Editor

PLOS Biology

cschnell@plos.org

REVIEWS:

Reviewer #1: The authors investigate a rare group of relatively circumscribed hippocampal lesion patients and characterise their behaviour in three conflict tests: an operant (approach-avoidance) conflict test, a Stroop task, and a go/nogo task. They find that patients with HC lesions approach more in the operant conflict test and are similar to healthy controls otherwise. 

The analyses are exhaustive and rigorous. From a neuropsychological perspective, this is an interesting data set. I have some concerns that the authors should address before publication.

1. Many statistical tests are reported but there appears to be no multiple comparison strategy - the p-values from these tests are therefore not interpretable in a strict sense. A statistically rigorous correction method should be implemented.

2. The results are framed as relevant for mental health disorders in a wider sense (i.e. beyond the HC lesion group). The idea that approach-avoidance conflict tasks are relevant for anxiety disorders is occasionally made in original research papers, but despite 7 decades of research there is no evidence for this assumption beyond the fact that GABA agonists reduce behavioural cautiousness in these tasks and also reduce subjective anxiety in patients (see for reviews e.g. (Cryan and Sweeney, 2011; Ennaceur, 2014; Ennaceur and Chazot, 2016; Harro, 2018; Bach, 2022). Approach-avoidance conflict is surely an important economic setup (i.e. a mixed gamble) with relation to real-life scenarios, but the clinical claim is probably unwarranted. 

3. It would be good if the authors could place their task within the wider space of human approach-avoidance conflict tests, which differ markedly.

4. Results are difficult to read. Perhaps some of the statistical minutiae could go into tables?

Bach DR (2022) Cross-species anxiety tests in psychiatry: pitfalls and promises. Mol Psychiatry 27:154-163.

Cryan JF, Sweeney FF (2011) The age of anxiety: role of animal models of anxiolytic action in drug discovery. Br J Pharmacol 164:1129-1161.

Ennaceur A (2014) Tests of unconditioned anxiety - pitfalls and disappointments. Physiol Behav 135:55-71.

Ennaceur A, Chazot PL (2016) Preclinical animal anxiety research - flaws and prejudices. Pharmacol Res Perspect 4:e00223.

Harro J (2018) Animals, anxiety, and anxiety disorders: How to measure anxiety in rodents and why. Behav Brain Res 352:81-93.

Reviewer #2: The study by W. Le Duc and colleagues investigates the contribution of the hippocampus to approach-avoid decision making in humans. Patients with circumscribed hippocampal damage and matched controls were evaluated on two versions of an automated approach-avoid conflict task administered on a monitor screen (one version of the task for objects and another one for scenes), as well as on two tasks evaluating response conflict (Stroop task and a cued Go/NoGo task). For the approach-avoid conflict task, the authors found that patients with hippocampal damage 'approached' conflict more often than controls. The authors applied a hierarchical drift diffusion model to examine group differences in the approach-avoid conflict task. For the critical conflict condition, they found that, relative to controls, patients with hippocampal damage were more biased to approach, required less evidence to make a decision, and were slower to accumulate evidence towards avoidance. In contrast, no group differences emerged in either of the two response conflict tasks.

This is an interesting study that reports novel and convincing evidence that hippocampal damage is associated with disruptions to latent decision-making processes during an approach-avoidance conflict task. The rationale for the study is clearly laid out, the tasks and procedures are clearly described, and the results adequately illustrated. The patients are well characterized with respect to their brain damage and neuropsychological status. The findings provide novel insight into the nature of the hippocampal contribution to approach-avoid decisions by showing that patients with hippocampal damage: 1) exhibited an increased willingness to approach; 2) showed virtually no accumulation of evidence toward avoidance; and 3) required less accumulated evidence to make decisions. The authors discuss the findings in the context of behavioral inhibition theory, with which the data are compatible. After exploring findings from human fMRI and electrophysiology and rodent lesion studies, they conclude that that 'behavioral inhibition' does not explain all the findings related to approach-avoidance paradigms. They suggest the novel possibility that hippocampal damage may disrupt representations of possible aversive outcomes. If so, the patients might be impaired or unable to even detect motivational conflict.

Specific recommendations are provided below. There is an issue with the notation used in describing the results of the hierarchical drift diffusion models (hDDMs), explained below. Despite this, it appears the authors' overall claims are supported. 

Specific comments: major

1) p. 3, Introduction, 2nd para: Either here or in the Discussion (or both) it would be appropriate to cite work in nonhuman primates that suggests hippocampus damage produces a greater propensity to approach reward in the presence of threat (Y. Chudasama et al., 2008).

2) Table 1. The value for Left HPC seems anomalous. How can this value be greater than the two values for aHPC and pHPC? Although the reader is directed to ref # 23 for details, it does not mention how HPC is calculated. Are aHPC and pHPC the same size as each other (so 'HPC' is simply the mean of aHPC and pHPC)? Please explain.

3) p. 17, Methods, end of 1st para: the authors describe the notation from their hierarchical Drift Diffusion Models (hDDMs) as though it can be read out as 'significant' or not. They are correct that capital P refers to the probability of overlap of the posterior distributions of two conditions, but I believe they are in error in assigning 'significance'. Interpreting the posterior probability (P) as a metric of statistical significance is antithetical to the Bayesian framework that underlies the hDDM. Moreover, this frequentist (mis)interpretation is particularly ill-advised because the hDDM does not always use the normal distribution (a key assumption in many Frequentist models) and instead utilizes non-normal distributions (e.g., Beta, Gamma) that are uncommon to most tests of statistical significance (see: Sequential Sampling Models — HDDM 0.9.8 documentation). It is more appropriate to interpret the P returned by the hDDM as the probability that the parameter estimate (e.g., Drift Rate) for one condition (e.g., Patients) is greater than that same parameter estimate from another condition. The authors should therefore remove mention of "significance" when discussing their hDDM results and instead interpret P values as the probability of superiority. Because the value of P will differ depending upon which posterior is used as the reference (see: Basic HDDM Tutorial — HDDM 0.9.8 documentation ), the authors should also edit their Results to explicitly mention which two posteriors are being compared. For example, p. 23 (1st sentence) could instead read as: "The control group drift rate was almost certainly lower/more negative relative to the patient group (P(Controls > Patients) = 0.0046) on [Conflict] trials..." 

Alternatively, if the authors prefer the Frequentist interpretation of statistical significance, they could remove all P values and instead use an estimation approach to hypothesis testing where the null hypothesis is rejected if the null value (e.g., 0) is not contained within the 95% highest density interval (HDI) of the posterior (see Kruschke, 2014).

4) p. 19, para 2 under Decision phase: The authors report cohen's d for all three comparisons combined (e.g., "all d ≥ 1.64, d 90% CI all [1.57 3.70]"), but this is unhelpful. Instead, authors should report all three Cohen's d effect sizes (and all 3 corresponding t values). Similarly, authors should refrain from reporting minimum effect sizes (e.g., b values from linear models) when the effects are statistically significant.

5) p. 22, Results, under Within-Group Comparisons: The authors should avoid interpreting P values as p-values (see Comment #3, above). Instead, authors should report that P(No-Conflict > Conflict)Patients= 0.49. The report that, "all P ≤ 0.018" is also misleading because P actually refers to the probability (X > Y). Please revise this notation throughout.

6) p. 22, Results, under Between-Groups Comparisons: As an example of the misuse of notation described in #3 (above), the authors state "On both No-Conflict Positive and No-Conflict Negative trials, there was no indication that groups differed significantly in estimates for either non-decision time (both Ps => 0.26), drift rate (both Ps => 0.77), or threshold (both Ps => 0.72)." This is confusing because the authors do not clarify which group is higher/lower than the other. Here and elsewhere, please explicitly state which parameters are being compared and which group shows the greater relative probability of superiority.

7) p. 29, first para: the idea regarding a role for the hippocampus in 'retrieval of evidence associated with undesirable outcomes' works for the present study and several others, but does not take into account the findings from studies in rodents and nonhuman primates that the conflict in approach-avoid conflict tasks can involve unlearned or innate threat (e.g., novel foods, snakes). For example, in rodent studies, ventral hippocampal damage disrupts innate avoidance of novel foods and entry onto open arms of a plus maze, which suggests a broader role for the hippocampus in resolving approach-avoid conflict. If so, this means the hippocampus is essential for not only conflict with recently learned items with conflicting valence, but also for some classes of innate ones. Indeed, it is worth discussing the point that the hippocampus plays an essential role not only in cases with learned 'cued' valences involving secondary reinforcers (points, as here) but also in cases in which approach-avoid conflicts appear to be based on innate (or genetically programmed) threats or rewards. These findings are more consistent with the authors' later statements (e.g., last sentence p. 29) that absence of a hippocampus may hamper the detection of motivational conflict.

Specific comments: minor

1) p. 12, Introduction, 2nd para: primate homolog of the *rodent* vHPC

2) p. 28, 2nd para; the authors state that the volume loss was 'predominantly within the anterior portion of the hippocampus'. From Table 2 it appears that the damage to aHPC was slightly greater than damage to the pHPC, but this does not strike me as 'predominantly' to aHPC. Please revise the statement or provide more information in support of the claim. Also, is it possible to run a correlation of the patient's approach scores against extent of aHPC damage? That might be more compelling. 

3) Figure 3: it is difficult to compare the amount of overlap in the Control and Patient plots in Fig 3 because the axes are different. Please make x-axis and y-axis the same for both plots within a panel. Also, the authors should consider separating the plots shown in Fig 3A-C (within-group comparisons) from those shown in D (between-group comparisons).

4) p. 28, second para: it would be appropriate to cite the work in nonhuman primates here (see # 1 under Specific comments: major).

5) p. 28, second para, last sentence: the word 'human' is used twice unnecessarily. Please revise.

6) p. 30, second para: the authors discuss a possible role for the perirhinal cortex in approach-avoid conflict. First, the fMRI study in humans that implicates perirhinal cortex is correlative, so not a direct conflict. Although they view the one rodent study as conflicting with the current study, it seems that an alternative explanation for the rodent findings is that the study relied on two objects (combined) in the approach-avoid scenarios. Because the perirhinal cortex is implicated in learning about the conjunctions of objects (Saksida et al., 2007), it is possible that this aspect of the studies was responsible for the effects of the perirhinal cortex manipulations. It may also be worth noting (here or elsewhere) that the anterior hippocampus has different anatomical connections than the posterior hippocampus, including direct projections to the prefrontal cortex (Barbas and Blatt, 1995).

References

Barbas H, and Blatt GJ (1995) Topographically specific hippocampal projections target functionally distinct prefrontal areas in the rhesus monkey. Hippocampus, 5: 511-533.

Chudasama Y, Wright KS and Murray EA (2008) Hippocampal lesions in rhesus monkeys disrupt emotional responses but not reinforcer devaluation effects. Biological Psychiatry, 63: 1084-1091.

Kruschke JK (2014) Doing Bayesian data analysis: A tutorial with R, JAGS, and Stan (2nd ed.). New York, NY: Academic/Elsevier.

Saksida LM, Bussey TJ, Buckmaster CA, and Murray EA (2007) Impairment and facilitation of transverse patterning after lesions of the perirhinal cortex and hippocampus, respectively. Cerebral Cortex, 17: 108-115.

---

## [Decision Letter · Decision Letter 2]

3 Dec 2024

Dear Dr Lee,

Thank you for your patience while we considered your revised manuscript "Hippocampal Damage Disrupts the Latent Decision-Making Processes Underlying Approach-Avoidance Conflict Processing" for consideration as a Research Article at PLOS Biology. Your revised study has now been evaluated by the PLOS Biology editors, the Academic Editor and the original reviewers. 

In light of the reviews, which you will find at the end of this email, we are pleased to offer you the opportunity to address the remaining points from the reviewers in a revision that we anticipate should not take you very long. We will then assess your revised manuscript and your response to the reviewers' comments with our Academic Editor aiming to avoid further rounds of peer-review, although might need to consult with the reviewers, depending on the nature of the revisions.

**IMPORTANT - SUBMITTING YOUR REVISION**

*Resubmission Checklist*

*Published Peer Review*

*PLOS Data Policy*

*Blot and Gel Data Policy*

Sincerely,

Christian

Christian Schnell, PhD

Senior Editor

PLOS Biology

cschnell@plos.org

REVIEWS:

Reviewer #1: Thank you for giving me the opportunity to review this revised manuscript. The authors have addressed my comments 2-4. However, my comment 1 remains to be addressed. The initial comment was: "Many statistical tests are reported but there appears to be no multiple comparison strategy - the p-values from these tests are therefore not interpretable in a strict sense. A statistically rigorous correction method should be implemented." The authors reply, in essence, that (a) presenting series of statistical tests without correction is the current norm in neurobiology, and (b) that their results survives multiple comparison correction and therefore their correction does not need to be reported. It is hard to find merit in these arguments.

1. It is a non sequitur to conclude from the prevalence of a reporting convention that it is rigorous or statistically valid. If journals and reviewers followed this argument then we would still be in the 1940s with no statistical tests being used in neurobiological research papers. Numerous papers continue to be published on how to improve statistical reporting, partly motivated by the recognition that a large part of research findings in life sciences does not replicate (see e.g. Ioannidis et al. 2005: Why most published research findings are false.)

2. The authors' second argument seems to be that multiple comparison correction is only necessary to report if results do not survive, but not if they do. This is out of step with conventions in statistical reporting: stastistical test methodology should be reported independent of the test result. Indeed, it is also in the interest of authors to report if their results are particularly strong, i.e. survive correction.

If the findings indeed survive correction for multiple comparison across the series of tasks and and number of response measures then this should simply be mentioned in the methods along the lines of "all p-values fell below the alpha level when correcting for the number of tasks and response measures, using the XY correction method." (preferably using Holm-Bonferroni -- not Bonferroni -- for XY). Otherwise, those results that did not survive should be flagged up.

Reviewer 2:

see attachment

---

## [Editor Report · Decision Letter 3]

13 Dec 2024

Dear Dr Lee,

Thank you for your patience while we considered your revised manuscript "Hippocampal Damage Disrupts the Latent Decision-Making Processes Underlying Approach-Avoidance Conflict Processing" for consideration as a Research Article at PLOS Biology. Your revised study has now been evaluated by the PLOS Biology editors and the Academic Editor. 

When assessing the revision, we noted that while you added a paragraph stating that most of your results survive correction for multiple comparisons, you have not actually updated the corrected statistics in all of the tables. Please update all the results and tables with the Bonferroni corrected stats, instead of leaving the uncorrected stats in the manuscript. Without this, we will not be able to proceed towards publication of your manuscript.

We will then assess your revised manuscript with our Academic Editor aiming to avoid further rounds of peer-review, although might need to consult with the reviewers, depending on the nature of the revisions.

**IMPORTANT - SUBMITTING YOUR REVISION**

*Resubmission Checklist*

*Published Peer Review*

*PLOS Data Policy*

*Blot and Gel Data Policy*

Sincerely,

Christian

Christian Schnell, PhD

Senior Editor

PLOS Biology

cschnell@plos.org

---

## [Editor Report · Decision Letter 4]

20 Dec 2024

Dear Andy,

Thank you for your patience while we considered your revised manuscript "Hippocampal Damage Disrupts the Latent Decision-Making Processes Underlying Approach-Avoidance Conflict Processing" for publication as a Research Article at PLOS Biology. This revised version of your manuscript has been evaluated by the PLOS Biology editors and the Academic Editor.

Based on our Academic Editor's assessment of your revision, we are likely to accept this manuscript for publication, provided you satisfactorily address the following data and other policy-related requests:

* We would like to suggest a different title to improve its accessibility for our broad audience: "Hippocampal Damage Disrupts the Latent Decision-Making Processes Underlying Approach-Avoidance Conflict Processing in humans"

* Please include information in the Methods section whether the study has been conducted according to the principles expressed in the Declaration of Helsinki.

* DATA POLICY:

Regardless of the method selected, please ensure that you provide the individual numerical values that underlie the summary data displayed in the following figure panels as they are essential for readers to assess your analysis and to reproduce it: 2CD and 5ABC.

* CODE POLICY

* In addition, current terminology should be used when referring to human participants and human participant groups/categories. Outmoded terms and potentially stigmatizing labels should not be used. Person-first language should be used in writing about people with diseases, disorders, or other conditions, except for communities that prefer identity-first terminology as their current standard. See https://www.nih.gov/nih-style-guide/person-first-destigmatizing-language for more information.

Examples:

* people with [name of disease, condition, or disorder]” should be used instead of terms such as “diabetics”, “cancer victims”, or “patients with”

We expect to receive your revised manuscript within three weeks. 

*Published Peer Review History*

*Press*

Kind regards and happy holidays,

Christian

Christian Schnell, PhD

Senior Editor

cschnell@plos.org

PLOS Biology

---

## [Editor Report · Decision Letter 5]

20 Jan 2025

Dear Andy,

Thank you for your patience while we considered your revised manuscript "Hippocampal Damage Disrupts the Latent Decision-Making Processes Underlying Approach-Avoidance Conflict Processing in Humans" for publication as a Research Article at PLOS Biology. 

As discussed, please replace “patients” by “people with hippocampal damage” in the abstract and throughout the manuscript. 

Please also make sure that the data are available in the repository you mention in the Data Availability Statement and remove those files from the submission. 

We expect to receive your revised manuscript within two weeks. 

*Published Peer Review History*

*Press*

Sincerely,

Christian

Christian Schnell, PhD

Senior Editor

cschnell@plos.org

PLOS Biology

---

## [Editor Report · Decision Letter 6]

23 Jan 2025

Dear Andy,

Thank you for the submission of your revised Research Article "Hippocampal Damage Disrupts the Latent Decision-Making Processes Underlying Approach-Avoidance Conflict Processing in Humans" for publication in PLOS Biology. On behalf of my colleagues and the Academic Editor, Raphael Kaplan, I am pleased to say that we can in principle accept your manuscript for publication, provided you address any remaining formatting and reporting issues. These will be detailed in an email you should receive within 2-3 business days from our colleagues in the journal operations team; no action is required from you until then. Please note that we will not be able to formally accept your manuscript and schedule it for publication until you have completed any requested changes.

PRESS

Sincerely, 

Christian

Christian Schnell, PhD

Senior Editor

PLOS Biology

cschnell@plos.org